JCB Journal of Cell Biology

**TOOLS**

# Cytosolic concentrations of actin binding proteins and the implications for in vivo F-actin turnover

Sofia Gonzalez Rodriguez[1]* , Alison C.E. Wirshing[1]* , Anya L. Goodman[1,2] , and Bruce L. Goode[1]

**Understanding how numerous actin-binding proteins (ABPs) work in concert to control the assembly, organization, and turnover of the actin cytoskeleton requires quantitative information about the levels of each component. Here, we measured the cellular concentrations of actin and the majority of the conserved ABPs in *Saccharomyces cerevisiae*, as well as the free (cytosolic) fractions of each ABP. The cellular concentration of actin is estimated to be 13.2 µM, with approximately two-thirds in the F-actin form and one-third in the G-actin form. Cellular concentrations of ABPs range from 12.4 to 0.85 µM (Tpm1> Pfy1> Cof1> Abp1> Srv2> Abp140> Tpm2> Aip1> Cap1/2> Crn1> Sac6> Twf1> Arp2/3> Scp1). The cytosolic fractions of all ABPs are unexpectedly high (0.6–0.9) and remain so throughout the cell cycle. Based on these numbers, we speculate that F-actin binding sites are limited in vivo, which leads to high cytosolic levels of ABPs, and in turn helps drive the rapid assembly and turnover of cellular F-actin structures.**

## Introduction

Actin networks are among the most complex molecular structures found in eukaryotic cells, with hundreds of moving parts that self-organize into elaborate force-generating networks (Pollard and Cooper, 2009; Blanchoin et al., 2014; Lappalainen et al., 2022). Understanding the formation and inner workings of these cellular assemblages requires having a complete molecular parts list of the actin-binding proteins (ABPs) and knowing their cellular abundances and their specific effects on actin filament dynamics and spatial organization. There is now a nearly complete inventory of the components of some F-actin structures, including the cortical patches, cables, and cytokinetic actin rings in budding yeast and fission yeast (Moseley and Goode, 2006; Kovar et al., 2011; Goode et al., 2015). In addition, for most of the conserved ABPs, their affinities for F-actin and each other have been determined, as well as their in vitro effects on actin filament nucleation, elongation, branching, cross-linking, severing, and depolymerization. Moreover, biochemical reconstitution studies have begun to define the cooperative and competitive relationships among specific pairs of ABPs, and how these relationships help sort ABPs into different actin networks found in vivo (Kadzik et al., 2020; Lappalainen et al., 2022).

Despite these advances, fundamental gaps in our knowledge remain which limit our ability to quantitatively describe and model how groups of ABPs work in concert to assemble, organize, and turnover F-actin networks. One of these gaps is the absence of quantitative information on the cytosolic ("free") concentrations of ABPs, which can profoundly influence an ABP's function in the cell, e.g., how rapidly an ABP binds to newly polymerized F-actin in vivo, and how mixtures of ABPs will compete and/or cooperate with each other in binding and regulating F-actin. The cytosolic concentrations of actin and some ABPs have been determined in fission yeast (Sirotkin et al., 2010) and in human neutrophils (Cano et al., 1992; DiNubile et al., 1995), but to our knowledge not in budding yeast.

Here, we have addressed this knowledge gap in the budding yeast *Saccharomyces cerevisiae*, which has served as an ideal model for dissecting actin regulation. *S. cerevisiae* has a relatively simple genome, with most ABPs encoded by a single gene, and almost all of its ABPs are conserved in structure and function across evolutionarily diverse organisms (Moseley and Goode, 2006; Goode et al., 2015). Another virtue of *S. cerevisiae* is that throughout most stages of the cell cycle, it has only two major F-actin structures, cortical patches and polarized cables. The patches are densely branched F-actin networks nucleated by the Arp2/3 complex which play an essential role in driving endocytosis. The cables are parallel bundles of actin filaments polymerized by formins, which serve as tracks for the transport of secretory vesicles and organelles and are essential for polarized cell growth. While patches and cables are strikingly different in their size and shape, both structures are highly dynamic. F-actin patches (~0.15 µm in diameter) polymerize at a rate of ~0.05 µm/s (~20 subunits/s), have a lifetime of 15–20 s, and turnover

[1]Department of Biology, Rosenstiel Basic Medical Science Research Center, Brandeis University, Waltham, MA, USA; [2]Department of Chemistry and Biochemistry, California Polytechnic State University SLO, San Luis Obispo, CA, USA.

*S. Gonzalez Rodriguez and A.C.E. Wirshing contributed equally to this paper. Correspondence to Bruce L. Goode: goode@brandeis.edu.



every ∼3 s (Kaksonen et al., 2003; Lacy et al., 2019). Thus, patches are thought to completely turnover five to six times during their relatively short lifetime at the cell cortex. Cables polymerize at a faster rate of ∼0.3–0.4 µm/s (∼100–150 subunits/s), but also grow much longer (5–7 µm) and turnover in ∼10–20 s (Yang and Pon, 2002; Yu et al., 2011; McInally et al., 2022).

S. cerevisiae actin patches and cables are decorated by distinct subsets of ABPs (Goode et al., 2015). Cables are decorated specifically by tropomyosins Tpm1 and Tpm2 (Liu and Bretscher, 1989; Pruyne et al., 1998). Patches are decorated specifically by Arp2/3 complex, Sac6/fimbrin, Abp1, capping protein, Crn1/coronin, twinfilin, and Scp1 (Drubin et al., 1988; Goode et al., 1998, 1999; Goodman et al., 2003). Abp140, which bundles F-actin, decorates both patches and cables (Asakura et al., 1998; Yang and Pon, 2002). Cables and patches both appear to be disassembled by cofilin, Aip1, and Srv2/CAP (Lappalainen et al., 1997; Rodal et al., 1999; Okada et al., 2006; Okreglak and Drubin, 2007, 2010; Chaudhry et al., 2013; Ydenberg et al., 2015; Pollard et al., 2020). While cofilin, Aip1, and Srv2/CAP are clearly visible on patches by immunofluorescence and GFP-tagging, these proteins are not readily detected on cables. Their absence from cables may possibly be explained by cables being less dense F-actin structures, making it difficult to detect proteins that sparsely decorate them. Alternatively, the actin filaments in patches may not disassemble as rapidly as those in cables upon binding of cofilin, Aip1, and Srv2/CAP. Importantly, cables are also decorated along their lengths by Tpm1 (and to a lesser degree Tpm2), which allows cables to grow long before being disassembled. These interactions with tropomyosin can also spatially restrict disassembly factors to the distal (pointed) ends of actin filaments (Jansen and Goode, 2019; Pollard et al., 2020). Consistent with this view, cables in tpm1∆ cells are very short and unable to support polarized growth or cell viability at elevated temperatures (Liu and Bretscher, 1989; Pruyne et al., 1998).

One of the next challenges is to gain a quantitative, systems-level view of how ABPs orchestrate the formation and turnover of actin structures like yeast patches and cables and to better understand how they simultaneously assemble such different structures side-by-side in a shared cytosol and sort different ABPs to each structure. As mentioned above, this requires knowing the cytosolic concentrations and molar ratios of actin and ABPs. Here, we have measured these numbers for actin and all 14 of the major ABPs found in S. cerevisiae using a combination of quantitative Western blotting and live imaging of fluorescently tagged proteins. We find that the cellular concentrations of ABPs vary greatly, which has important implications for their functions and how they work together. Furthermore, our data reveal that the sum concentration of ABPs greatly exceeds the concentration of actin in cells, which allows cells to maintain high cytosolic concentrations of ABPs, and in turn drive rapid actin assembly and turnover.

## Results

### The cellular levels of *S. cerevisiae* actin and actin-binding proteins

We began by measuring the total cellular abundances of actin and 14 ABPs in S. cerevisiae. For actin and 11 of the ABPs, we used quantitative Western blotting and for the remaining three ABPs, we used GFP-intensity analysis. Western blotting required raising polyclonal antibodies to purify S. cerevisiae actin and eight of the ABPs and to optimize antibody dilutions for blotting (Table S1). On immunoblots, signals were compared between different amounts of cell lysate and known quantities of purified protein (Fig. S1 A). This yielded the abundances of each protein in ng per µg of total cellular protein, from which we calculated the molar abundance in cells (Table 1). Using the molar abundance and the cytoplasmic volume of a haploid yeast cell (Fig. S1 B), we derived the concentration of each protein in the cytoplasm (see equation, Fig. 1 A). Note, we define "cytoplasm" here as the internal volume of the cell, excluding the volume occupied by the cell wall and major organelles (more details in Materials and methods).

Because we did not have reliable antibodies for Tpm2, Pfy1, and Abp140, we calculated their total cellular concentrations using alternative strategies. For Tpm2, we started with the concentration of endogenous (untagged) Tpm1 determined by Western blotting above (12.4 µM) and then estimated Tpm2 concentration based on the observation that Tpm2 and Tpm1 are present in yeast cells at a 1:6 molar ratio (Drees et al., 1995). For Pfy1 and Abp140, we tagged each protein with GFP and compared the total fluorescent signal of the fusion protein (Pfy1-GFP, Abp140-GFP) in cells to standard curves from other ABPs tagged with the same fluorescent protein, for which we had their abundance from quantitative Western blotting (details in Fig. S1 C and Materials and methods). Since it has been difficult historically to make a functional Pfy1-GFP strain, we tested the growth properties of our Pfy1-GFP strain at 25, 34, and 37°C (Fig. S1 D). Our Pfy1-GFP strain grew similar to WT at each temperature, suggesting that the GFP fusion is functional.

Our results indicate that the total cellular concentration of actin (F-actin plus G-actin) in the cytoplasm of S. cerevisiae is 13.2 ± 2.44 µM (mean ± SD) and that the concentrations of ABPs range from 0.85 to 12.4 µM (Fig. 1 B). We found it useful to also display these data in a landscape view, where the area of each circle in the diagram is proportional to the cellular concentration of the protein, and the lines between proteins represent well-supported physical interactions (Fig. 1 C). This landscape view highlights the relative abundance of each ABP to actin and other ABPs. Interestingly, two of the most abundant ABPs are Tpm1 (12.4 µM) and cofilin (7.58 µM), which have implications for their known antagonistic relationship in cable formation (see Discussion).

We were also interested in determining how the cellular pool of actin in S. cerevisiae (13.2 µM) is distributed between filamentous (F-actin) and globular (G-actin) forms. For vertebrate cells, one method commonly used is the biochemical separation of the triton soluble (G-actin) and insoluble (F-actin) fractions after cell lysis (Koestler et al., 2009). However, all of the actin in yeast depolymerizes immediately upon cell lysis (Goode, 2002), precluding the use of this approach. Another method relies on GFP tagging of actin; however, GFP-actin fusions are nonfunctional in yeast and GFP-actin is not incorporated into cables by formins (Doyle and Botstein, 1996). Since methods for directly measuring the G-actin-to-F-actin ratio in S. cerevisiae are

Table 1. **Quantitative summary of yeast actin-binding proteins**

| *S. cerevisiae* name | Common name | MW (kD) | Cellular abundance (ng/µg)[a] | n[b] | Cellular concentration (µM)[c] | Number of molecules per cell[d] | % free in cytosol[e] | Cytosolic concentration (µM)[f] |
|---|---|---|---|---|---|---|---|---|
| Act1 | Actin | 41.8 | 7.99 ± 1.48 | 11 | 13.2 ± 2.44 | $2.17 \times 10^5$ | - | - |
| Tpm1 | Tropomyosin | 23.5 | 4.22 ± 1.08 | 6 | 12.4 ± 3.16 | $2.03 \times 10^5$ | 89 ± 8 | 11 ± 0.98 |
| Pfy1[g] | Profilin | 13.6 | 1.54 ± 0.27 | - | 7.73 ± 1.38 | $1.27 \times 10^5$ | - | - |
| Cof1 | Cofilin | 16 | 1.76 ± 0.78 | 3 | 7.58 ± 3.36 | $1.25 \times 10^5$ | - | - |
| Abp1 | Actin-binding protein 1 | 65.6 | 4.91 ± 0.79 | 2 | 5.15 ± 0.83 | $8.47 \times 10^4$ | 61 ± 11 | 3.2 ± 0.60 |
| Srv2/CAP | Cyclase-associated protein | 57.6 | 2.41 ± 0.36 | 5 | 2.88 ± 0.43 | $4.74 \times 10^4$ | 78 ± 9 | 2.2 ± 0.26 |
| Abp140[g] | Actin binding protein | 71.4 | 2.94 | - | 2.83 ± 0.60 | $4.65 \times 10^4$ | 87 ± 8 | 2.5 ± 0.23 |
| Tpm2[h] | Tropomyosin | 19.1 | 0.70 | - | 2.54 | $4.17 \times 10^4$ | 91 ± 8 | 2.3 ± 0.20 |
| Aip1 | Actin-interacting protein 1 | 67.4 | 1.81 ± 0.02 | 2 | 1.85 ± 0.02 | $2.40 \times 10^4$ | 82 ± 7 | 1.5 ± 0.13 |
| Cap1/2 | Capping protein (α/β dimer) | 64.3 | 1.43 ± 0.53 | 7 | 1.46 ± 0.54 | $5.12 \times 10^4$ | 67 ± 8 | 0.9 ± 0.15 |
| Crn1 | Coronin | 72.6 | 1.48 ± 0.28 | 6 | 1.40 ± 0.27 | $2.31 \times 10^4$ | 63 ± 11 | 0.9 ± 0.15 |
| Sac6 | Fimbrin | 71.8 | 1.26 ± 0.02 | 2 | 1.21 ± 0.02 | $1.99 \times 10^4$ | 80 ± 8 | 1.0 ± 0.10 |
| Twf1 | Twinfilin | 37.1 | 0.59 ± 0.19 | 8 | 1.10 ± 0.35 | $1.80 \times 10^4$ | 71 ± 12 | 0.8 ± 0.13 |
| Arp2 | Actin-related protein 2 | 44.1 | 0.63 ± 0.20 | 4 | 0.98 ± 0.31 | $1.62 \times 10^4$ | 81 ± 7 | 0.8 ± 0.07 |
| Scp1 | Calponin/transgelin | 22.8 | 0.28 ± 0.05 | 3 | 0.85 ± 0.15 | $1.39 \times 10^4$ | 74 ± 13 | 0.6 ± 0.11 |

Proteins are listed in order from highest to lowest cellular concentration.

[a]Abundance (mean ± SD) is expressed as ng of ABP per µg of total cellular protein determined by quantitative Western blotting as in Fig. S1.

[b]n, number of times cellular abundance was determined by quantitative Western blotting.

[c]Cellular concentrations of proteins were calculated as in Fig. 1 A and Materials and methods.

[d]Number of molecules per cell calculated from the cellular concentration of each protein and the average cytosolic volume of a yeast cell, calculated in Fig. S1.

[e]Percentage of ABP in the cytosol (mean ± SD) was calculated by live imaging as shown in Fig. 2.

[f]Cytosolic concentrations of ABPs; data from Fig. 3 C.

[g]Concentration of these ABPs were determined based on fluorescence of GFP fusions rather than Western blot (details in Materials and methods).

[h]Concentration based on previous study (Drees et al., 1995) showing that Tpm2 is approximately sixfold less abundant than Tpm1 in cells.

currently not available, we estimated the concentration of G-actin by relying on three observations: (i) the rate of cable polymerization in vivo of ~0.3 µm/s (Yang and Pon, 2002; Huckaba et al., 2006; Eskin et al., 2016; McInally et al., 2022), (ii) 370 actin subunits/µm of filament (Hanson and Lowy, 1963), and (iii) the rate of filament elongation in vitro by *S. cerevisiae* formins Bni1 and Bnr1 of ~25 subunits s[−1] µM[−1] using profilin-actin (Chesarone-Cataldo et al., 2011; Graziano et al., 2011). From these numbers, we calculated that the cytosol would need to contain ~4.40 µM profilin-G-actin to support this rate of cable growth (see Materials and methods). Subtracting the cytosolic concentration of G-actin (4.4 µM) from the total cellular concentration of actin (13.2 µM), we arrive at an estimation for the concentration of F-actin in cells (8.8 µM). These are of course only estimates, and once new methods are developed to directly measure F-actin and/or G-actin levels in yeast, these numbers can and should be revisited. However, even as rough estimates, these values can be useful for discussing the physiological implications of the ABP concentrations measured.

Interestingly, our data show that the sum concentration for all of the ABPs that bind F-actin (Tpm1, Cof1, Abp1, Srv2, Abp140, Tpm2, Aip1, Cap1/2, Crn1, Sac6, Twf1, Arp2/3, and Scp1) is 42.5 µM, which greatly exceeds the estimated 8.8 µM F-actin in cells (Fig. 1 D). This observation suggests that binding sites on F-actin may be limiting in vivo and predicts that a large fraction of the ABPs should be in the cytosol (a hypothesis we confirm below). Similarly, the sum concentration of ABPs that bind G-actin (Cof1, Pfy1, Srv2, and Twf1) is 19.3 µM, which greatly exceeds the estimated 4.4 µM G-actin in cells (Fig. 1 E). This observation suggests that most of the actin monomers in cells are bound to ABPs and that there may be very little free (unbound) G-actin in the cytosol.

Since we used two distinct methods to determine total cellular levels of ABPs, Western blotting and GFP fluorescence intensity, we were also interested in comparing the values

Figure 1. **Cellular concentrations of S. cerevisiae actin and ABPs. (A)** Equation used to calculate cellular concentrations of actin and ABPs (example shown is for actin). The starting value, actin abundance in cells (ng/μg) comes from quantitative immunoblotting (Fig. 1, A and B; and Table 1). The μg of total protein in a haploid *S. cerevisiae* cell is known (Johnston et al., 1977; Baroni et al., 1989). The cytosolic volume was calculated in Fig. S1 B. **(B)** Graph comparing the total cellular concentrations of actin plus 14 ABPs (mean ± SD). Abundances of actin and 11 of the ABPs were determined by quantitative immunoblotting as described in A. Asterisks denote the other three ABPs (Pfy1, Tpm2, and Abp140), for which antibodies were not available for immunoblotting. Therefore, the total cellular concentrations of these three proteins were calculated by GFP-tagging and comparison to GFP-tagged ABPs of known quantity (see Materials and methods). The sample size for each Western blot in descending order (excluding proteins marked with an asterisk): 11, 6, 3, 2, 5, 2, 7, 6, 2, 8, 4, 3 (also listed in Table 1). **(C)** Landscape view of *S. cerevisiae* actin and ABP quantities and their interactions. The area of each circle is proportional to the cellular concentration of each protein, and lines connecting circles represent known direct interactions (Moseley and Goode, 2006; Goode et al., 2015). **(D)** Graph comparing the relative abundance of cellular actin in filamentous form (F-actin) to the sum abundance of the F-actin-binding ABPs. **(E)** Graph comparing the relative abundance of cellular actin in monomeric/globular form (G-actin) to the sum abundance of the G-actin-binding ABPs.

obtained by these two approaches (Fig. S2 A). Overall, similar values were obtained by these methods, particularly for Abp1 (5.15 versus 5.55 μM), Crn1 (1.40 versus 1.56 μM), and Cap2 (1.46 versus 1.62 μM). An exception was Sac6, which was 1.21 μM by Western blotting and 7.71 μM by fluorescence intensity. Importantly, while the Sac6-GFP fusion complements function

in vivo (Kim et al., 2006), it is possible that the GFP tag retards the normal turnover of Sac6 in vivo, resulting in abnormally high levels of Sac6-GFP without compromising function.

We also compared our results from Western blotting and GFP intensity to results from a study integrating multiple *S. cerevisiae* proteomic mass spectrometry data sets (Ho et al., 2018; Fig. S2

B). Our Western blotting data agree well with the mass spectrometry data for actin (13.2 versus 14.8 µM, respectively), Srv2/CAP (2.88 versus 2.69 µM), Sac6 (1.21 versus 1.40 µM), and Scp1 (0.85 versus 1.02 µM). However, our GFP intensity data do not agree well with the mass spectrometry data, except in the case of Twf1 (0.53 versus 0.65 µM). The better overall agreement between our Western blotting data and the mass spectrometry data suggests that GFP intensity (with tagged proteins) may not be as reliable a method for defining ABP cellular concentrations. For this reason, in Fig. 1, we show the cellular concentrations of ABPs determined by Western blotting, and we only use the GFP intensity values for those ABPs for which we did not have Western blotting data.

## Quantifying the distributions of *S. cerevisiae* ABPs between the cytosol and F-actin structures

Having measured the total cellular concentrations of actin and ABPs, we next wanted to determine the concentration of each ABP that is available ("free") in the cytosol to regulate new patches and cables. This information can be quite useful because the rate at which an ABP binds to newly polymerized actin scales linearly with its free concentration. Therefore, the free concentrations of ABPs can influence both the order and extent to which different ABPs decorate newly polymerized F-actin and exert their regulatory effects on actin dynamics and/or spatial organization.

To measure the cytosolic concentrations of ABPs, we generated yeast strains with chromosomally integrated GFP tags on each ABP (at the endogenous locus) along with an integrated Arc15-mScarlet to mark patches or a plasmid expressing Lifeact-mScarlet to mark cables. Importantly, ABP-GFP levels in cells were similar in the presence and absence of the Arc15-mScarlet marker (Fig. S3). Using dual-color live cell imaging, we acquired z-stacks through the entire volume of the cell and from these generated a 2D intensity projection (examples in Fig. 2, A and B). For each cell, we also generated a mask of its F-actin patches or cables and used this to measure the intensity of the ABP-GFP signal in the "unmasked region." Dividing this value by the ABP-GFP intensity for the entire cell (no masking) yielded the fraction of ABP-GFP in the cytosol. Importantly, the masking was not used to measure ABP-GFP signal in F-actin structures (patches and cables), but rather to exclude them from our measurement of mean ABP-GFP intensity in the remaining areas (cytosol). Therefore, as long as the signal from the F-actin structures is completely excluded, the thresholding used to generate the F-actin masks should not affect our quantification of cytosolic concentrations. We also found that increasing or decreasing the threshold for generating the F-actin masks had little effect on our measurements of the ABP cytosolic concentrations (Fig. 2 C).

By this strategy, we analyzed cells expressing 11 different GFP-tagged ABPs and 3 different mNeon-tagged ABPs. Cof1 was excluded from this analysis because integrated GFP fusions failed to localize to patches like untagged Cof1 or to complement cell growth and actin organization in vivo. In contrast, mNeon-tagged Tpm1 and Tpm2 localized properly to cables, similar to untagged Tpm1 and Tpm2, and therefore were included in the analysis. We note however, that mNeon-Tpm1 and mNeon-

Tpm2 were expressed over the endogenous copies of Tpm1 and Tpm2 because on their own they did not fully complement the function.

We observed a wide range of fluorescence intensities among ABPs due to their different expression levels and cytosolic fractions (examples in top row, Fig. 3 A). As expected, Pfy1-GFP was highly abundant and appeared entirely in the cytosol. mNeon-Tpm1 was also highly abundant and showed high cytosolic levels. The fluorescence distributions of ABPs between the cytosol and F-actin structures varied for different GFP-tagged ABPs (examples in bottom row, Fig. 3 A). Quantitative image analysis revealed that 61–91% of each ABP is in the cytosol (Fig. 3 B). Similar results were obtained for Cap1-GFP and Cap2-GFP strains, as expected, given they are stoichiometric subunits of the same complex. The high cytosolic levels of the ABPs agree well with our earlier prediction that F-actin binding sites are limited based on comparing the sum concentration of F-actin binding ABPs to the concentration of F-actin in cells (Fig. 1 D). From the cytosolic fractions of each ABP and their cellular concentrations (Fig. 1 and Table 1), we calculated the free cytosolic concentrations (not F-actin bound), which ranged from 0.6 to 11 µM (Fig. 3 C).

We also examined cell-to-cell variance in the total cellular levels of ABP-GFPs by plotting the raw intensities for each individual cell analyzed (Fig. S4 A). In addition, since the mean intensities vary considerably among ABP-GFPs, we replotted the data after normalizing the means of the ABP-GFPs to each other, which allowed a more accurate comparison of the cell-to-cell variance (Fig. 4 B). Finally, we also calculated the coefficient of variation (CV), which is the standard deviation divided by the mean, for each ABP-GFP (Fig. S4 C). Together, these analyses show that for most ABP-GFPs, the cell-to-cell variance is very low (CV = 0.18–0.37). It is slightly higher for two ABPs, Scp1-GFP (CV = 0.51) and mNeon-Tpm2 (CV = 0.42). This may stem from their GFP signals being more faint, which can lead to a higher variance after background subtraction.

## Cytosolic levels of ABPs remain constant throughout the cell cycle

Yeast cells change dramatically in size and shape over the cell cycle. They start as unbudded cells in G1, then polarize and progress through small-, medium-, and large-budded stages, finally undergoing cytokinesis to yield unbudded mother and daughter cells. These morphological changes are accompanied by dynamic rearrangements of the actin cytoskeleton (Moseley and Goode, 2006). Actin cables and patches are initially randomly organized in unbudded cells, then rapidly polarize upon initiation of bud emergence. Throughout bud formation, cables run parallel to the mother-bud axis, extending from the bud tip and bud neck, while patches are concentrated at the bud cortex in a corral surrounding the site of exocytosis directed by cables. Leading up to cytokinesis, patches and cables briefly resume a random organization, then repolarize at the bud neck, with cables extending into both the mother and bud compartments. A cytokinetic actin ring transiently forms to facilitate septation, and, finally, the actin ring is disassembled upon completion of cytokinesis.

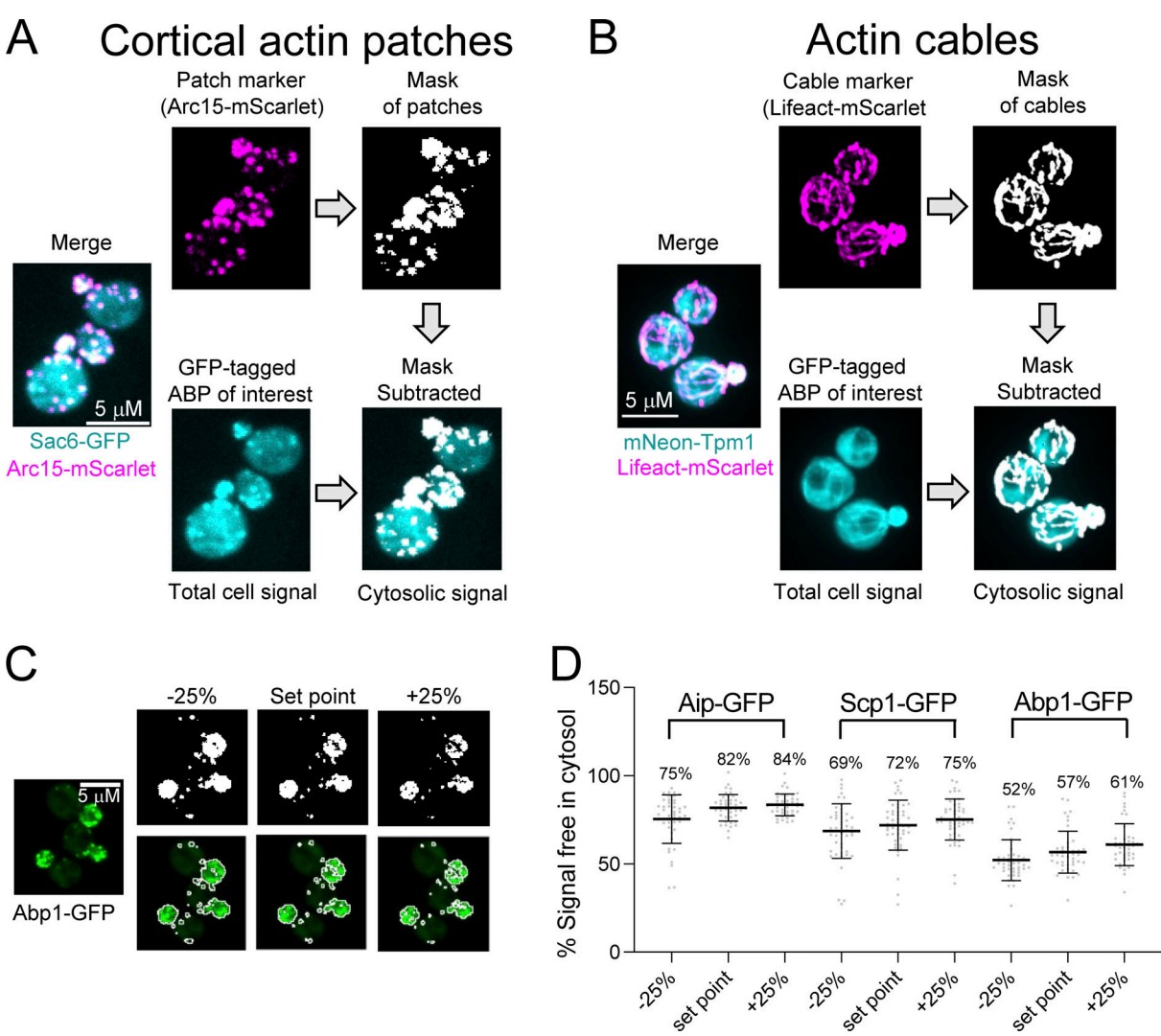

Figure 2. **Strategy for quantifying distributions of ABPs between cytosol and cellular F-actin structures. (A)** Scheme for ABPs that localize to F-actin patches. Yeast strains co-expressing an actin patch marker (Arc15-mScarlet) along a GFP-tag on the ABP of interest were imaged by confocal microscopy, taking optical sections through the entire cell. The mean fluorescence intensity of each ABP-GFP fusion was quantified for each cell. Then the patch marker was used to generate a "mask," which was superimposed on the ABP-GFP image. The "unmasked" region was defined as the cytosol and its mean intensity was measured and used to calculate the fraction of the total cellular ABP-GFP signal in the cytosol. **(B)** Scheme for ABPs that localize to actin cables. Same as in A, except strains used coexpress a cable marker (Lifeact-mScarlet) along with an ABP of interest tagged with mNeon. **(C)** Large changes in the threshold value used to make F-actin masks lead to only small changes in the percentage of GFP signal in the cytosol. Shown are patch masks obtained using three different threshold values overlaid on the ABP-GFP image. The "set point" (center) is the threshold value used for generating masks of the patch ABPs analyzed in this study. Masks on either side of the set point were generated by decreasing (left) or increasing (right) the set point value by 25%. **(D)** Graph comparing the percent ABP-GFP signal in the cytosol obtained after analyzing the same cell images using the three different threshold values in C. Each data point represents one cell. Total number of cells analyzed for each ABP (from left to right): 43, 49, 44. Three ABPs were selected for this analysis because they show a range in cytosolic fraction, from relatively high (Aip1-GFP), to medium (Scp1-GFP), to relatively low (Abp1-GFP).

Given these dramatic changes in cell morphogenesis and actin organization, we asked whether the cytosolic levels of specific ABPs rise or fall during the cell cycle. To address this, we reanalyzed our data on the cytosolic fractions of each ABP (Fig. 3) as a function of cell area. Remarkably, cytosolic levels of all 14 ABPs remained consistent over the cell cycle (Fig. 4). Similarly, the total cellular levels of ABPs remained constant over the cell cycle (Fig. S5). These observations agree with proteomic studies showing that most yeast proteins are maintained at a steady concentration in vivo even as cell volume increases (Lin and Amir, 2018). Overall, these observations raise

the possibility that the expression levels of yeast ABPs have been optimized for maintaining high cytosolic concentrations of each ABP, which in turn has implications for how this ensemble of ABPs drives rapid rearrangements in actin organization seen at different stages of the cell cycle.

## Discussion

In this study, we have measured the total cellular (cytoplasmic) concentrations, free (cytosolic) concentrations, and molar ratios of actin and 14 conserved ABPs in *S. cerevisiae*. Combining these

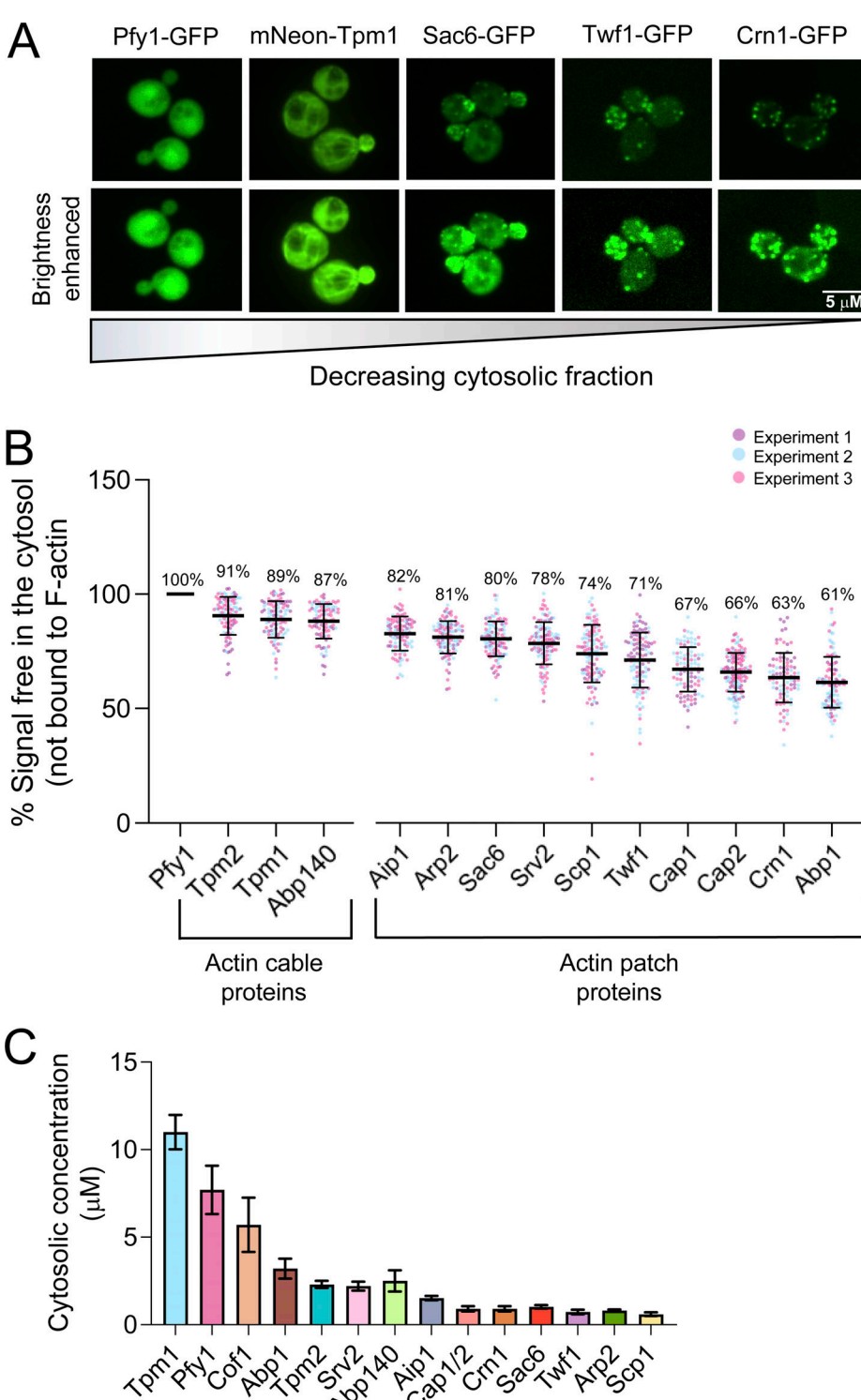

Figure 3. **Distributions of ABPs between the cytosol and F-actin structures. (A)** Representative average intensity projections from confocal imaging of cells expressing fluorescently tagged patch or cable ABPs. The top panel shows cells expressing fluorescently tagged ABPs with a range of different intensities, and due to their different cellular abundances; images were acquired and processed identically. Bottom panel shows the same cell images but with the brightness adjusted to improve visualization of the fluorescence distribution between cytosol and F-actin structures. Note these strains also express either a patch marker (Arc15-mScarlet) or a cable marker (Lifeact-mScarlet) for generating an F-actin mask (see strategy in Fig. 2). **(B)** Percentage of GFP signal in the cytosol (not bound to F-actin) for each ABP (mean ± SD). Data are from three independent experiments (n > 20 cells per experiment). Each data point represents one cell. Total number of cells analyzed for each ABP (left to right): 96, 98, 95, 96, 102, 116, 112, 122, 113, 99, 137, 92, 112. **(C)** Cytosolic concentrations (mean ± SD) of each ABP in descending order, calculated by multiplying the cellular concentration of each ABP by its fraction in the cytosol (Table 1). Cof1 is marked by an asterisk; its cytosolic concentration was estimated (rather than measured) from its total cellular concentration, making the assumption that its cytosolic fraction is within the range of other ABPs (61–91%).

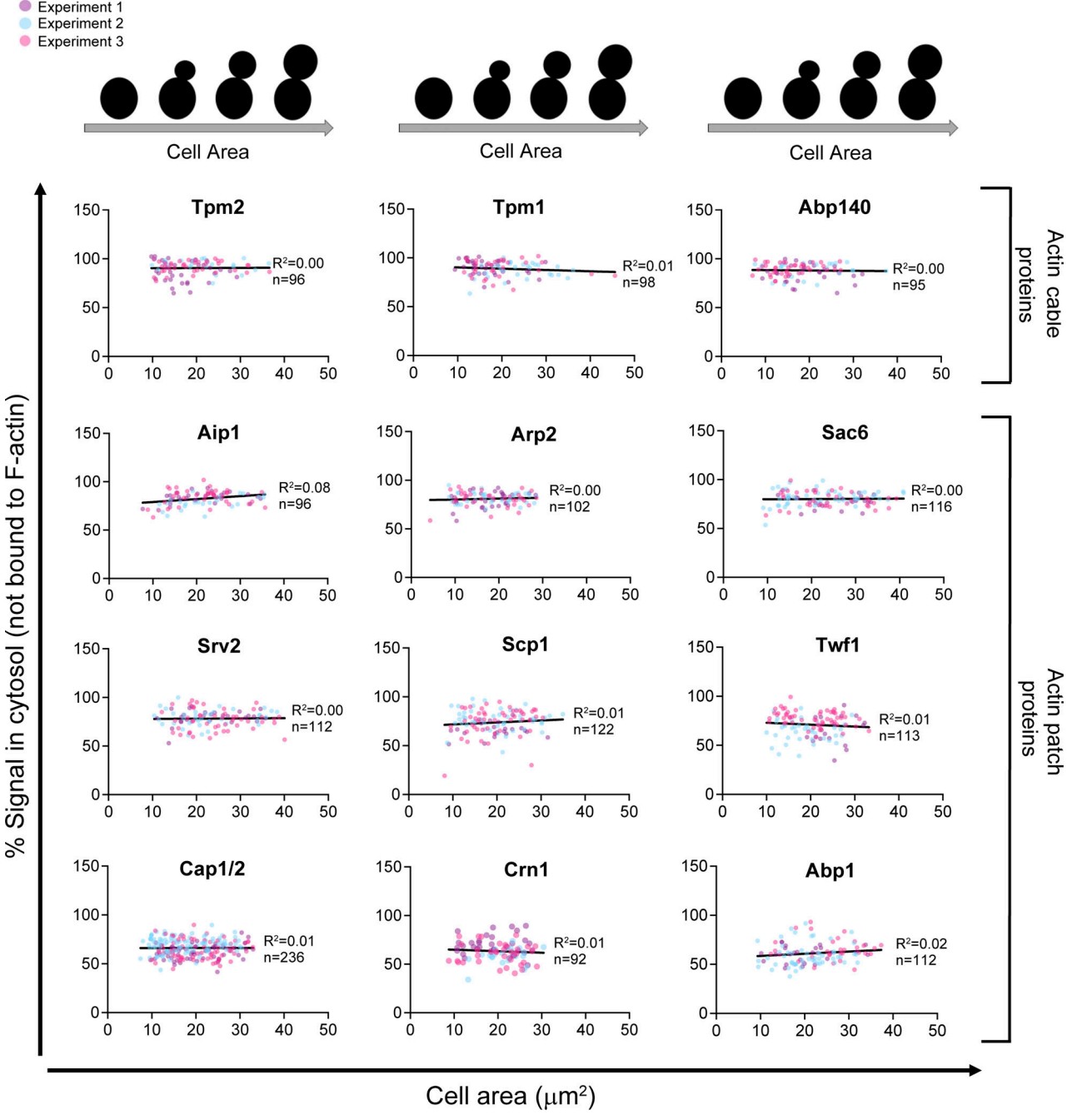

Figure 4. **Cytosolic concentrations of ABPs show no major fluctuations over the cell cycle.** For each ABP, the data from Fig. 3 B, showing the percentage of ABP in the cytosol for each individual cell, was replotted against the area (μm²) of that cell. Linear regression analysis was used to correlate cytosolic concentration of ABP with cell area (μm²) within the population.

numbers with the known interactions and activities of these proteins, we can begin to build a more quantitative systems-level view of how this group of proteins works in concert, co-operatively and competitively, to control the assembly and turnover of F-actin patches and cables (Fig. 5). Cytosolic concentrations can be even more useful than total cellular concentrations because they represent the concentrations of ABPs in cells available to bind and regulate newly polymerized F-actin. When combined with the on-rates of ABPs for F-actin, cytosolic concentrations can inform the order and extent to which

different ABPs bind to filaments, which strongly influences actin dynamics and spatial organization.

**Impact of high cytosolic concentrations of ABPs on the rapid turnover of yeast F-actin structures**
Patches and cables are short-lived structures, with lifetimes of 15–20 s. Furthermore, recent high-resolution single molecule imaging work from Berro and colleagues has demonstrated that the actin subunits and ABPs in a patch turnover every ~3 s (Lacy et al., 2019), making patches one of the most dynamic F-actin

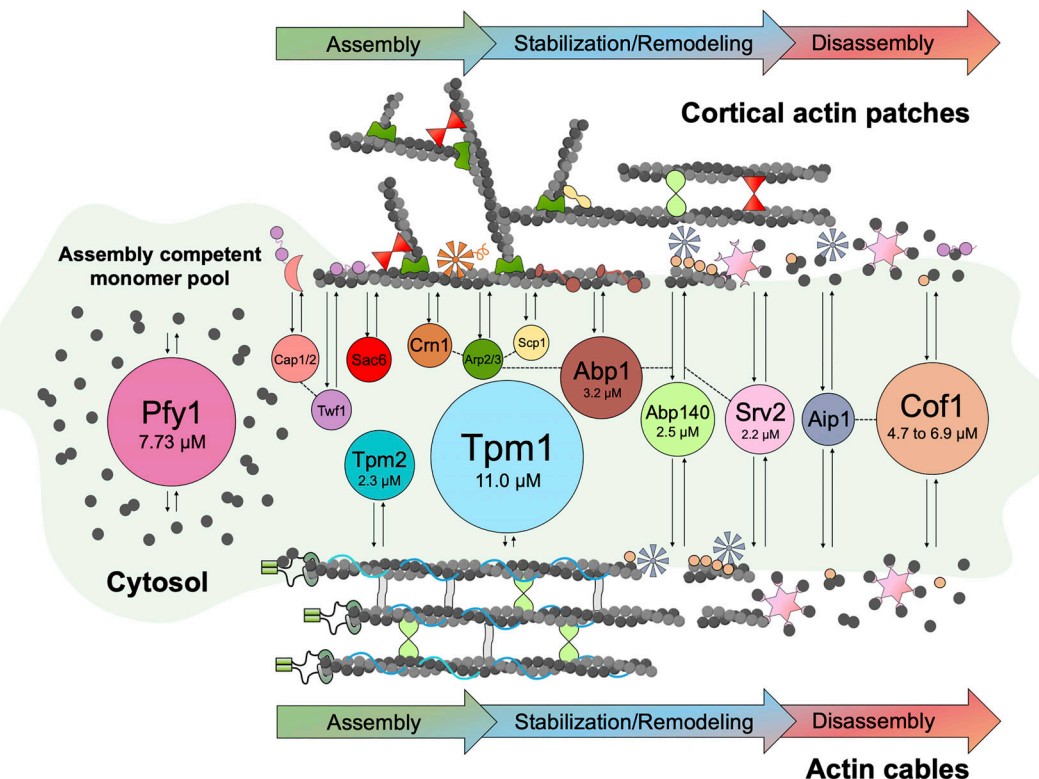

**Figure 5.** **Cartoon showing cytosolic concentrations of ABPs available to regulate the formation and turnover of F-actin patches and cables.** Cartoon depicting cytosolic ABPs (circles) and their interactions with cellular F-actin structures: cortical patches (above) and cables (below). ABPs have been spatially organized to highlight their known effects on F-actin assembly, organization, remodeling, and disassembly. The area of each ABP circle is proportional to its cytosolic concentration listed in Table 1. Cytosolic ABPs are color-coded to match ABPs decorating F-actin networks (with shapes that reflect their known structures). Actin cables are shown decorated by bundlers (light gray) of unknown identity. Bidirectional arrows indicate the rapid exchange of ABPs between F-actin structures and cytosol (Lacy et al., 2019). Dashed lines between circles represent known physical interactions. Concentrations of select ABPs are shown as points of reference. For Cof1, we could not directly measure its cytosolic fraction, and instead it was estimated operating under the assumption that it falls within the range of the other ABPs (61–91% in the cytosol). Since profilin (Pfy1) binds specifically to G-actin, it is shown interacting with the assembly-competent actin monomer pool available for assembly of patches and cables. Note that Srv2, Twf1, and Cof1 interact with F-actin as well as G-actin, as depicted.

structures known in any organism. This predicts that all of the molecular components of a patch turnover five to six times during the patch's short lifetime. It also means that only 1–2 s after actin subunits are added to a patch they are already being targeted for disassembly.

What is the physiological benefit of having such fast F-actin turnover in yeast? It has been suggested that the rapid assembly and turnover of the actin cables allow yeast cells to respond rapidly to internal and external cues and redirect membrane traffic and polarity (Moseley and Goode, 2006; Dyer et al., 2013). In addition, recent modeling and simulation studies suggest that the rapid turnover of actin patches enables these networks to generate persistent forces that deform the invaginating endocytic membrane and continuously dissipate elastic stress while maintaining network form and integrity (Hiraiwa and Salbreux, 2016; McFadden et al., 2017). How then are rapid actin turnover dynamics maintained in vivo? Berro and colleagues have theorized that rapid patch turnover should require ABPs to decorate F-actin very shortly after its polymerization, likely within ~0.05 s (Mousavi et al., 2022 *Preprint*). The high cytosolic concentrations of ABPs we have measured here support fast rates of

binding. For instance, the cytosolic concentrations of Abp1 (3.2 µM) and Srv2 (2.2 µM), assuming diffusion-limited rates of binding of ~10 s$^{-1}$ µM$^{-1}$, would lead to binding to new sites on F-actin within 0.05 s of polymerization. In principle, the extremely high cytosolic concentration of Tpm1 (11 µM) may lead to even faster decoration of cables. Since formins and profilin together polymerize F-actin more than five times faster than polymerization at patches, the high cytosolic concentration of Tpm1 may ensure its decoration of cables, protecting the growing cables from other ABPs. This would allow cables to grow long before eventually being disassembled by cofilin, Aip1, and Srv2. While binding of Tpm1 does not block Tpm2 and Abp140 decoration of cables, it is known to block other ABPs (e.g., Arp2/3 complex, fimbrin, and cofilin) and may help "steer" them to patches (Fig. 5; Bernstein and Ramburg, 1982; Blanchoin et al., 2001; Bugyi et al., 2010; Skau and Kovar, 2010; Christensen et al., 2017).

Given that rapid actin turnover requires high concentrations of ABPs available to bind actin, how then is this free pool of ABPs maintained? Interestingly, we found that the sum concentration of F-actin-binding ABPs far exceeds the concentration of F-actin

in *S. cerevisiae* (Fig. 1 D). Thus, the high concentrations of ABPs in the cytosol may result from limiting binding sites on F-actin. This suggests that the expression level of actin in *S. cerevisiae* may be optimized to limit and maintain an excess of free ABPs in the cytosol. Consistent with this view, one study found that doubling the level of actin in *S. cerevisiae* is lethal (Binkley, 2001), possibly because it reduces the concentrations of free ABPs.

Comparing our numbers to the levels of actin and five of the same ABPs in *S. pombe* (Sirotkin et al., 2010), we see remarkable consistency in the high cytosolic fractions of these five ABPs (Table S2) despite *S. pombe* and *S. cerevisiae* being as evolutionarily distant from each other as they are from mammals (Sipiczki, 2000). On the other hand, the cellular concentrations of most of these ABPs is higher in *S. pombe* than *S. cerevisiae*, possibly related to the higher concentration of actin in *S. pombe* (22 µM) compared to *S. cerevisiae* (13.2 ± 2.44 µM). Thus, ABP levels may scale with actin levels, such that they maintain a high fraction of ABPs in the cytosol. In vertebrate cells, there are even higher concentrations of actin: ~100 µM G-actin and >100 µM F-actin (Mogilner and Edelstein-Keshet, 2002; Koestler et al., 2009). However, there are also hundreds of ABPs, many of which are expressed as multiple isoforms (Meenakshi S et al., 2023), and therefore the sum concentration of ABPs may again exceed the actin levels, leading to high fractions of ABPs in the cytosol. It is also important to note that in any given cell type, there may be considerable variation in the molar ratios of different ABPs to each other (e.g., note the differences between *S. cerevisiae* and *S. pombe*). This variability may reflect the distinct evolutionary paths taken by each organism, as well as differences in the types of actin networks assembled by each organism and cell type.

### New insights into the relationships between Tpm1 and Cof1 in shaping actin networks

Our analysis reveals that the concentrations of ABPs in yeast range almost 20-fold, from 0.6 to 11 µM. This quantitative information should be helpful to researchers studying the relationships among specific subsets of these proteins. Here, we discuss one example. We show that two of the most abundant ABPs are Cof1 and Tpm1, which is interesting because they are known to have a highly antagonistic relationship on actin filament sides and in vivo (Skau and Kovar, 2010; Schmidt et al., 2015; Christensen et al., 2017). At patches, Cof1 (assisted by Aip1 and Srv2) drives the rapid severing and depolymerization of F-actin, and on cables, Tpm1 protects the sides of filaments from Cof1 and its co-factors, delaying cable disassembly so that the individual filaments in cables can grow much longer than the filaments in patches. Consistent with this view, in *tpm1Δ* cells, cables are extremely short and fail to support polarized cell growth at elevated temperatures; however, these defects can be rescued by *aip1Δ* or *srv2Δ* mutations (Okada et al., 2006; Chaudhry et al., 2013). Taking these previous genetic observations together with the numbers we have measured here, we propose that high levels of Cof1 are key to driving the rapid turnover of actin patches. However, the presence of high levels of Cof1 in yeast cells in turn necessitates maintaining high levels

of Tpm1 to rapidly decorate cables and allow them to grow long in an actin disassembly-promoting environment. In other words, such high levels of Tpm1 would not be necessary for cable formation if not for the high levels of cofilin and its co-factors. One question that remains open and deserves future attention is how Tpm1 and Cof1 (both present at high levels in the cytosol) are specifically sorted to actin cables and patches, respectively. On the one hand, it has been demonstrated that yeast tropomyosin competes with cofilin and fimbrin for binding actin filaments and that the competition among these proteins is important for their sorting to different actin networks (Skau and Kovar, 2010; Christensen et al., 2017, 2019). On the other hand, it is still not clear why tropomyosin specifically decorates actin cables, while fimbrin and cofilin decorate actin patches.

### The actin monomer pool

Our data show that the cytosol in *S. cerevisiae* contains relatively high levels of Pfy1, Cof1, Srv2, and Twf1 (7.7, 5.8, 2.2, and 0.8 µM, respectively), the sum of which is approximately four times greater than the estimated concentration of G-actin. If one considers these high concentrations together with the high affinities of Cof1, Pfy1, Srv2, and Twf1 for G-actin, it suggests that the majority of actin monomers in yeast cells are bound to ABPs, and there is little if any free G-actin. Further, when combined with other available information on these proteins (below), it suggests a specific order of binding among these ABPs during the actin monomer recycling process (from ADP- to ATP-bound states). Cof1, Twf1, Pfy1, and Srv2 all compete for binding G-actin, and Srv2 catalyzes the dissociation of Cof1 and Twf1 from ADP-actin monomers (Mattila et al., 2004; Chaudhry et al., 2010; Johnston et al., 2015; Kotila et al., 2018). Among these four ABPs, Pfy1 has the highest affinity for ATP-G-actin ($K_d$ = 100 nM), while Srv2 has the highest affinity for ADP-G-actin ($K_d$ = 18 nM) but has 100-fold lower affinity for ATP-G-actin ($K_d$ = 2 µM). Assuming that the major products of F-actin disassembly in vivo are ADP-actin monomers bound to Cof1 or Twf1, we propose that high cytosolic concentrations of Srv2 lead to rapid displacement of Cof1 and Twf1 from ADP-actin monomers, followed by Srv2-catalyzed nucleotide exchange (ATP for ADP) on monomers, and a handoff of ATP-actin monomers to Pfy1. This model is consistent with earlier biochemical studies comparing the roles of yeast Srv2 and profilin in nucleotide exchange (Chaudhry et al., 2010, 2014), and with more recent studies that reconstitute actin-based motility in a closed system revealing that Srv2/CAP is critical for recycling monomers (Colin et al., 2022 *Preprint*).

We have estimated that the yeast cytosol contains 4.4 µM assembly-competent G-actin bound to profilin. How is this reservoir of G-actin maintained at a level ~30 times higher than the critical concentration for F-actin assembly? The answer is likely to involve several factors, including high levels of profilin to bind actin monomers, high levels of capping protein to rapidly bind free barbed ends of actin filaments, and tight restriction of in vivo actin nucleation events. Collectively, these mechanisms are predicted to limit the availability of barbed ends, leading to a "funneling effect," in which actin monomers accumulate to high levels in the cytosol until they are added in bursts upon the formation of new barbed ends by nucleation events (Carlier and

Shekhar, 2017). Rapid F-actin disassembly and monomer recycling ensure that this large pool of ATP-G-actin is maintained in the cytosol.

## Concluding remarks

In summary, the numbers we have measured here help elevate our quantitative understanding of how actin networks are assembled and turned over in vivo. Integrating the key discussion points from above, we propose that a combination of (i) limited actin expression, (ii) high cytosolic levels of capping protein, Tpm1, and actin disassembly factors, and (iii) tightly restricted actin nucleation events leads to an accumulation of G-actin in the cytosol (~4.4 µM), which in turn drives the rapid assembly of patches and cables. At the same time, an abundance of ABPs, relative to F-actin, maintains high concentrations of ABPs in the cytosol, which promotes the rapid decoration and turnover of F-actin patches and cables. Ultimately, the levels of actin and ABPs must be measured individually for each system. When more numbers become available for different organisms and cell types, it should be possible to compare systems and see the different ways they are wired, which ABPs are the most variable, and how the differences are compensated for by altered expression of the remaining ABPs.

## Limitations of this study

Since Lifeact is a peptide derived from the actin-binding domain of Abp140, it is likely to compete with Abp140. Therefore, our estimate of Abp140-mNeon cytosolic levels may be artificially high, as Lifeact-mScarlet was expressed in these cells to create a mask of F-actin (as in Fig. 2 B). The counterargument is that the cytosolic fraction we measured for Abp140-mNeon was similar to other cable-associated ABPs (Tpm1 and Tpm2), so the presence of the Lifeact marker may not cause a major change in the distribution of Abp140-mNeon between cytosol and F-actin structures. Cof1 and Twf1 each bind to F-actin and G-actin, making it difficult to know what fraction of these ABPs is truly "free" in the cytosol versus bound to G-actin, and thus the effective (free) cytosolic concentrations of Cof1 and Twf1 may be lower than what is reported here. Importantly, Srv2 is not subject to the same limitation because it binds to F-actin and G-actin via distinct domains. It is also important to keep in mind that for any given ABP, a significant portion of the "free" cytosolic pool may be inactive and/or unable to bind F-actin due to posttranslational modifications and/or inhibitory ligands. As this information becomes available for each ABP, it must be factored into any models assembled from the numbers we provide here. We expressed the concentration of capping protein as the concentration of the heterodimer, as it is comprised of two different polypeptides (Cap1 and Cap2). All other ABPs were expressed as the concentrations of their monomeric polypeptides, to facilitate thinking about their competitive relationships and how much actin each can bind. However, some of these ABPs oligomerize (e.g., Srv2 and Crn1), and therefore, one should keep in mind how this will affect their on-rates for F-actin and other ligands. For instance, Srv2 forms hexamers in vivo (Balcer et al., 2003), and therefore ~3 µM Srv2 polypeptide represents ~0.5 µM Srv2 hexamers capable of binding

to F-actin and/or G-actin. This will affect calculations of the concentration-dependent rates of Srv2 binding to actin. Conversely, each Srv2 polypeptide in a hexamer can bind to one actin monomer and one profilin, so for other calculations involving stoichiometries or percent occupancy (e.g., Fig. 1 E), it may be more relevant to consider the concentration of Srv2 polypeptide rather than hexamers. It is known that some ABPs can be recruited to actin patches in vivo independent of their direct physical interactions with F-actin. For instance, twinfilin can be recruited through its interactions with capping protein and Srv2/CAP can be recruited through its interactions with Abp1. While this does not change our conclusions about what fraction of these proteins are cytosolic, it needs to be considered when modeling competition for binding sites on F-actin in vivo.

# Materials and methods

## Yeast strains

A complete list of yeast strains used in this study is provided in Table S3. For analyzing the in vivo localization patterns of the following ABPs found on actin patches (Abp1, Aip1, Arp2, Cap1, Cap2, Crn1, Sac6, Scp1, Srv2, and Twf1), we started with ResGen yeast strains (Thermo Fisher Scientific) already carrying integrated C-terminal GFP tags on each ABP (Longtine et al., 1998; Huh et al., 2003). We then used PCR-based homology-directed repair (Longtine et al., 1998) to integrate an mScarlet tag at the C-terminus of the ARC15 gene in these strains. PCR products were amplified from a pFA6A-mScarlet-KAN vector using primers with 40-bp homology arms as described (Longtine et al., 1998). Tagging Arc15 with mScarlet in these ABP-GFP strains allowed the identification and masking of patches. For analyzing the localization of ABPs that decorate actin cables (Tpm1, Tpm2, and Abp140), we used existing strains with integrated N-terminal (Tpm1 and Tpm2) or C-terminal (Abp140) mNeon tags (Wirshing et al., 2023) and further transformed these strains with a 2-µm plasmid expressing Lifeact-mScarlet (#168085; Addgene) for identification and masking of the cables (see Fig. 2 B). Note that mNeon-Tpm1 was ectopically integrated at the LEU2 locus, and mNeon-Tpm2 at the TRP1 locus, and expressed over endogenous (untagged) Tpm1 and Tpm2. With the exception of the Lifeact-mScarlet cable marker, which is expressed from a plasmid, all other proteins were tagged directly in the genome (integrated) and expressed from their native promoters as the sole source of that protein in the cell.

## Antibody production and quantitative Western blotting

For this study, eight new polyclonal antibodies were generated in chickens against purified S. cerevisiae actin, Abp1, Aip1, Cap1/Cap2, Cof1, Srv2, Sac6, and Tpm1 (Aves Inc.). Only the Tpm1 and Aip1 antibodies required further affinity-purification for specificity (Harlow and Lane, 1988). All antibodies and their sources are listed in Table S1, including dilutions used for immunoblotting. For quantitative immunoblotting, total cellular extracts were prepared from a wild-type yeast strain (BGY12) grown in rich medium (YPD) to an $OD_{600} = 1$. Cultures were centrifuged at

10,000 × *g* for 5 min, the supernatant was aspirated, and total protein extracts were prepared (Horvath and Riezman, 1994) with minor modifications as follows. 200 μl of 0.2 M NaOH was added to 2 OD units of pelleted cells and incubated for 10 min at room temperature. Cells were pelleted, supernatant aspirated, and 1 vol of cell pellet was added to 2 vol of boiling SDS sample buffer (20 mM Tris-HCl [pH 6.8], 5% glycerol, 1% SDS, 7.5% β-mercaptoethanol, and 0.0012% bromophenol blue). After boiling for 3 min, samples were centrifuged at 10,000 × *g* for 10 min and stored at –20°C in small aliquots until used for immunoblotting. For quantification of Aip1 levels, an additional TCA precipitation step was introduced after the NaOH incubation, which was necessary to reliably preserve Aip1 abundance (actin levels were the same with both protocols). Total protein concentration in extracts was determined by Coomassie spot assay (Minamide and Bamburg, 1990) using a BSA standard curve. For each protein analyzed by Western blotting, three different quantities of total cell extract were loaded along with known quantities of purified protein. Primary antibody dilutions used to probe blots are listed in Table S1. HRP-conjugated secondary antibodies were used at a 1:10,000 dilution: anti-chicken (Aves, Inc.), and anti-mouse and anti-rabbit (GE Healthcare). Blots were developed with Supersignal West Pico Chemiluminescent substrate (Pierce) and signals were quantified by densitometry. Bands with intensities in the linear range of the standard curve were used to calculate the amount (ng) of each ABP per amount (μg) of total cellular protein loaded, yielding the cellular abundance of each protein in ng/μg (Table 1).

To calculate the cellular concentration of each ABP (Table 1), we used the relative abundances (ng ABP/ μg total cellular protein) of each ABP obtained by immunoblotting, and the following information: (i) an average haploid *S. cerevisiae* cell contains ∼4 × 10⁻¹² g of total protein (Johnston et al., 1977; Baroni et al., 1989), (ii) the density of a yeast cell is ∼1.1126 *g*/ml (Baldwin and Kubitschek, 1984), (iii) the wet weight of an average haploid yeast cell is ∼6 × 10⁻¹¹ g (Sherman, 2002), and (iv) ∼50% of the *S. cerevisiae* cell volume is the cytoplasm, while the remaining 50% is occupied by the cell wall and organelles (nucleus, vacuole, mitochondria, ER, Golgi, peroxisomes, etc.; Uchida et al., 2011) were obtained. The calculations performed using these values are outlined in Fig. 1 A, which yielded the molar concentration of each ABP in the cytoplasmic fraction of cells (includes cytosol and F-actin structures in the cell but excludes cell wall and organelles; Table 1). All proteins were expressed as concentrations of the monomeric polypeptide in cells. The concentration of capping protein (Cap1/Cap2) was expressed as the concentration of heterodimers.

**Fluorescence live-cell imaging**

To estimate the total cellular concentrations of Tpm2, Pfy1, and Abp140 (Fig. 1, B and C), for which we did not have antibodies (precluding quantitative Western blotting), we used alternative approaches. For Tpm2, we relied on a previous study showing that Tpm2 is present at one-sixth of the molar abundance of Tpm1 in yeast cells (Drees et al., 1995) and divided the Tpm1 concentration by six to derive the Tpm2 concentration. For Pfy1

and Abp140, we tagged the endogenous proteins with GFP, measured the mean GFP fluorescence intensity in cells (n > 80), as described in detail below, and compared the average total fluorescence signal in cells to a standard curve generated with other GFP-ABPs for which we had quantitative Western blotting data. The same method was used to determine ABP-GFP cellular concentrations shown in Fig. 2, A and B, comparing them to ABP cellular concentrations determined by Western blotting.

To quantify the fraction of each ABP in the cytosol (i.e., not bound to F-actin) versus those associated with F-actin structures (patches or cables; Figs. 2 and 3), we used a two-color live imaging approach. Strains co-expressing a specific ABP (tagged with GFP or mNeon) and either Arc15-mScarlet or Lifeact-mScarlet were grown in liquid cultures YPD media (2% glucose) at 25°C to OD₆₀₀ 0.3–0.6. Cells were pelleted, washed twice in synthetic complete media, and mounted on 5% agarose pads made with synthetic complete media for live imaging. Visualization of patch-associated ABPs (and Pfy1-GFP) was performed at room temperature on an i-E upright confocal microscope (Nikon Instruments) with a CSU-W1 spinning disk head (Yokogawa), 100× oil objective (NA 1.4; Nikon Instruments), and an Ixon 897 Ultra-CCD camera (Andor Technology) controlled by NIS-Elements software (Nikon Instruments). Exposure times of 200 ms at 50% laser power (excitation 488 and 561 nm) were used to acquire 35 Z-slices (at 0.2 μm steps) spanning the entire volume of the cell. To visualize cable-associated ABPs (Tpm1, Tpm2, and Abp140), we used super-resolution microscopy, which improved our ability to identify and mask individual cables. Imaging was performed at room temperature on a Nikon ECLIPSE Ti2 inverted microscope equipped with a Yokogawa CSU-W1 spinning disk with a SoRa module for super-resolution imaging, 60× (NA 1.4) and 100× (NA 1.45) oil immersion objectives, a Teledyne Photometrics Prime BSI Express Scientific CMOS (sCMOS) camera, and a Nikon LUN-F laser launch with four solid state lasers: 405, 488, 561, 640 nm. Exposure times of 100 ms at 50% laser power (excitation 488 nm) and 50 ms at 50% laser power (excitation 561 nm) were used to acquire 35 Z-slices (at 0.2 μm steps) spanning the entire volume of the cell. Cell images were acquired and denoised using Nikon Elements software.

All image analysis was done in ImageJ (Schindelin et al., 2012). We first generated an average intensity projection of the 488 channel to measure mean GFP or mNeon intensity fluorescence in the cell (average intensity per unit area). Then a cellular segmentation software, Cellpose, was used to generate whole cell outlines (Stringer et al., 2021). Signal in the 488 channel from autofluorescence was measured for cells not expressing GFP or mNeon and subtracted from all measurements of cells expressing GFP- or mNeon-tagged ABPs. The Arc15-mScarlet or Lifeact-mScarlet signal (from a maximum intensity projection of the entire cell) was used to generate a mask of actin patches or cables, respectively. The mask was overlaid on the average intensity projection of the ABP-GFP or ABP-mNeon image to exclude patches or cables and measure the mean GFP-ABP fluorescence intensity in the cytosol (average fluorescence per unit area). The mean GFP or mNeon fluorescence intensity in the cytosol (unmasked region) was divided by the mean GFP

or mNeon fluorescence intensity in the whole cell, yielding the fraction of ABP-GFP or ABP-mNeon signal in the cytosol.

## Statistical analysis

All experiments were repeated at least three times and yielded similar results. Means and errors (SD or SEM, as indicated in legends) were calculated using GraphPad Prism, as were linear regression. Data distributions were assumed to be normal, but this was not formally tested.

## Online supplemental material

Fig. S1 provides an example standard curve from Western blot analysis for quantifying the cellular abundance of actin, as well as the calculation used to determine cytosolic volume, the standard curves for calculating Pfy1-GFP and Abp140-GFP concentrations, and demonstration that Pfy1-GFP complements function in vivo (all related to Fig. 1). Fig. S2 compares the cellular concentrations of ABPs we determined by Western blotting versus ABP-GFP fluorescence measurements and compares these levels to ABP levels determined in previous mass spectrometry studies (related to Fig. 1). Fig. 3 shows that cellular levels of GFP-tagged ABPs are similar in yeast strains with or without an additional Arc15-mScarlet marker (related to Fig. 3). Fig. S4 uses coefficient of variance analysis to show that there is low cell-to-cell variation in the fluorescence levels of GFP-tagged ABPs. Fig. S5 shows that the cellular concentrations of the ABPs do not change significantly over the cell cycle. Table S1 provides information on each antibody used for Western blot analysis. Table S2 compares the free (cytosolic) concentrations of ABPs determined in this study in the budding yeast *S. cerevisiae* to those determined previously by Sirotkin and colleagues in the fission yeast *S. pombe*. Table S3 provides the genotypes of the *S. cerevisiae* strains used in this study.

## Data availability

Data are available from the corresponding author upon reasonable request.

## Acknowledgments

We are grateful to Amity Manning and Aaron Skolnik for technical assistance in the early stages of this work.

This research was supported by grants from the National Institutes of Health to A.C.E. Wirshing (F32 GM135967) and B.L. Goode (R35 GM134895).

Author contributions: Conceptualization, review, and editing manuscript—S. Gonzalez Rodriguez, A.C.E. Wirshing, and B.L. Goode. Data curation, investigation, methodology, visualization, validation, and formal analysis—S. Gonzalez Rodriguez and A.C.E. Wirshing. Drafting of manuscript—S. Gonzalez Rodriguez and B.L. Goode. Project administration, supervision, and resources—B.L. Goode.

Disclosures: The authors declare no competing interests exist.

Submitted: 7 June 2023

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

# Supplemental material

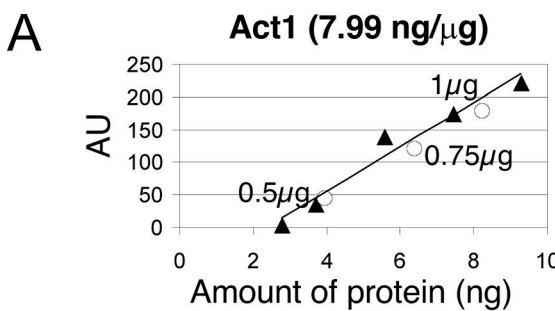

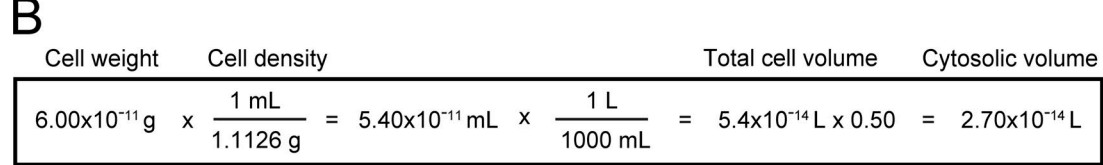

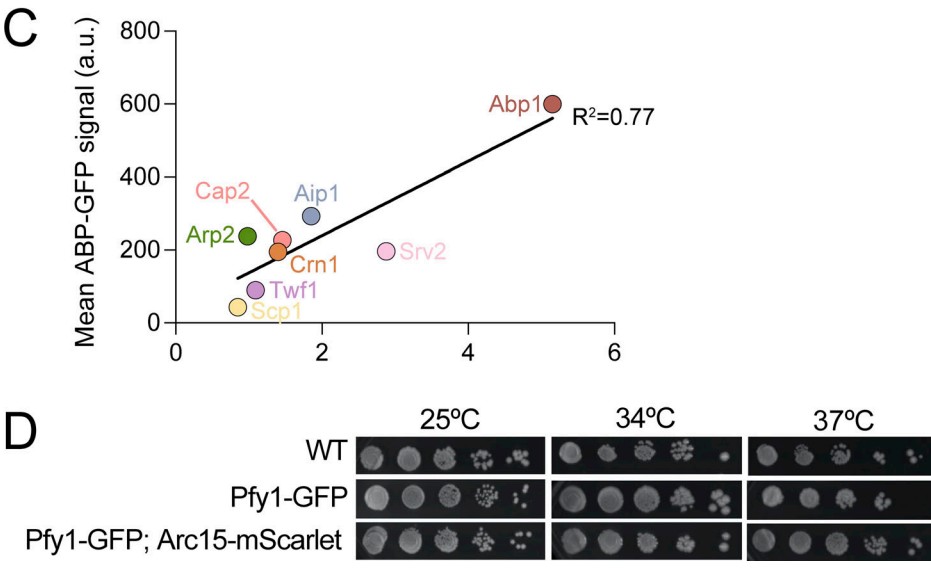

Figure S1. **Strategy for quantifying cellular concentrations of _S. cerevisiae_ ABPs. (A)** Example standard curve for quantifying the cellular abundance of actin. Signals (AU) on immunoblots for known quantities of purified actin were plotted (filled triangles) to generate a standard curve. Signals from different quantities of total cell lysate on the same immunoblot were plotted (open circles) to determine the amount of actin in the lysates (ng per μg total protein). **(B)** Calculation of the cytosolic volume of a haploid _S. cerevisiae_ cell using known values for the weight and density of a haploid yeast cell (Baldwin and Kubitschek, 1984; Sherman, 2002). **(C)** Standard curve used to estimate the cellular concentrations of Pfy1-GFP and Abp140-GFP, since we did not have reliable antibodies for Pfy1 or Abp140. For eight ABPs, confocal microscopy was used to determine total ABP-GFP concentrations in cells. These values were plotted against cellular concentrations determined by quantitative Western blotting (as in B; also see Materials and methods). Linear regression was used to generate the line shown, which has an equation (y = 102× +35.22) from which we estimated the cellular concentrations of Pfy1 and Abp140. **(D)** A C-terminal GFP tag on profilin (Pfy1) does not compromise growth at the indicated temperatures. Serial dilutions comparing three strains (WT, Pfy1-GFP, and Pfy1-GFP Arc15-mScarlet) grown on YPD plates for 48 h at 25°C, 34°C, and 37°C.

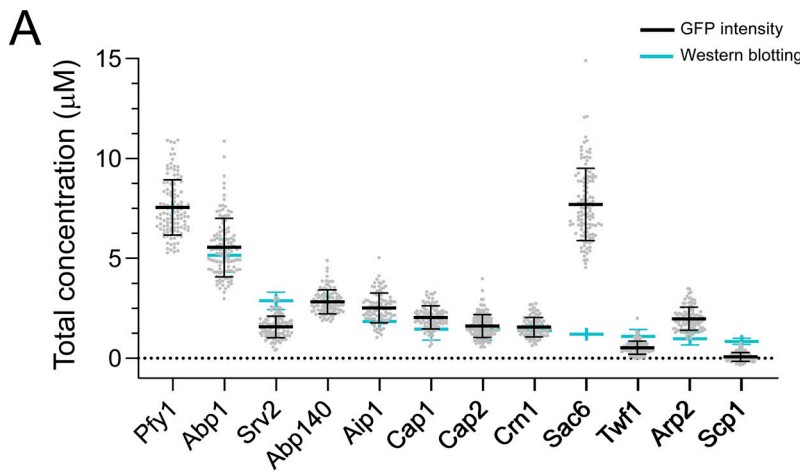

| | Cellular Concentration (μM) Western blotting This study | Cellular Concentration (μM) GFP intensity This study | Cellular Concentration (μM) Mass spectrometry Ho et al., 2018* |
|---|---|---|---|
| Act1 | 13.2 ± 2.44 | -- | 14.8 |
| Tpm1 | 12.4 ± 3.16 | -- | 3.58 |
| Pfy1 | -- | 7.55 ± 1.38 | 1.92 |
| Cof1 | 7.58 ± 3.36 | -- | 23.7 |
| Abp1 | 5.15 ± 0.83 | 5.55 ± 1.46 | 1.08 |
| Srv2/CAP | 2.88 ± 0.43 | 1.58 ± 0.54 | 2.69 |
| Abp140 | -- | 2.83± 0.64 | 0.43 |
| Tpm2 | -- | 2.54 | 2.70 |
| Aip1 | 1.85 ± 0.02 | 2.52 ± 0.74 | 0.73 |
| Cap1 | 1.46 ± 0.54 | 2.05 ± 0.58 | 1.81 |
| Cap2 | 1.46 ± 0.54 | 1.62 ± 0.57 | 0.57 |
| Crn1 | 1.40 ± 0.27 | 1.56 ± 0.49 | 0.73 |
| Sac6 | 1.21 ± 0.02 | 7.71 ± 1.81 | 1.40 |
| Twf1 | 1.10 ± 0.35 | 0.53 ± 0.33 | 0.65 |
| Arp2 | 0.98 ± 0.31 | 1.98 ± 0.58 | -- |
| Scp1 | 0.85 ± 0.15 | 0.07 ± 0.22 | 1.02 |

*In Ho et al., 2018 the mass spectrometry data was presented as a number of molecules per cell. For comparative purposes, we converted that to the cellular concentration (μM) using the cell volume shown in Supplemental Figure 1C.

Figure S2.   **Comparison of total ABP cellular concentrations quantified by Western blotting versus GFP fluorescence. (A)** Graph comparing the total cellular concentrations of 12 different ABPs. Western blot data (mean ± SD) are from multiple experiments (as listed in Table 1 and Fig. 1 B). GFP data shown (mean ± SD) are from three independent experiments (*n* > 20 cells per experiment), with each data point representing one cell. GFP intensity values were converted to cellular concentrations (μM) as in Fig. S1 D. **(B)** Table comparing ABP cellular concentrations determined by Western blotting (this study), GFP fluorescence (this study), and mass spectrometry (Ho et al., 2018).

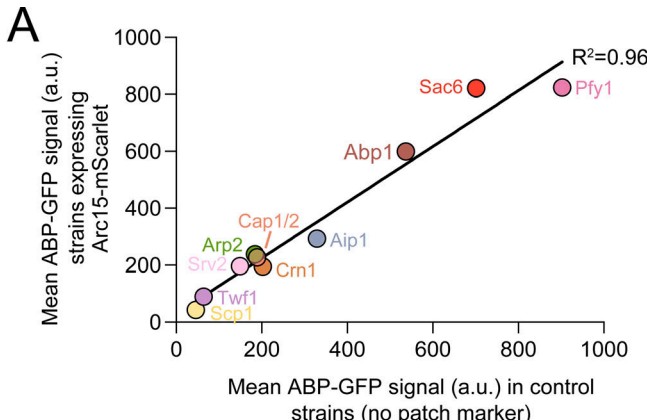

| | Mean ABP-GFP signal (a.u.) | | | |
|---|---|---|---|---|
| | Strains **not** expressing Arc15-mScarlet | n | Strains **expressing** Arc15-mScarlet | n |
| Pfy1 | 902.7 | 36 | 823.7 | 117 |
| Sac6 | 701.2 | 44 | 821.3 | 116: |
| Abp1 | 536.8 | 39 | 599.4 | 98 |
| Aip1 | 328.9 | 44 | 292.8 | 96 |
| Crn1 | 202.3 | 47 | 194.7 | 92 |
| Cap1/2 | 189.4 | 40 | 227.0 | 236 |
| Arp2 | 184.3 | 47 | 237.6: | 102 |
| Srv2 | 149.0 | 39 | 196.2 | 112 |
| Twf1 | 64.3 | 58 | 89.4 | 113 |
| Scp1 | 45.6 | 37 | 42.6 | 122 |

Figure S3.   **Correlation between cellular levels of GFP-tagged ABPs in yeast strains with or without additional Arc15-mScarlet marker. (A)** Graph comparing the mean GFP fluorescence intensity values for yeast strains co-expressing Arc15-mScarlet and the GFP-tagged ABP (y-axis) and yeast strains expressing only the GFP-tagged ABP (x-axis). For each cell, an average intensity projection was generated, giving the mean GFP fluorescence intensity for that cell. Data from all cells were averaged to give the values graphed (circles). For strains co-expressing Arc15-mScarlet ($n > 80$ cells per condition) and for strains not expressing Arc15-mScarlet ($n > 35$ cells per condition). **(B)** Table showing the average GFP fluorescence values for the cells and the number of cells analyzed.

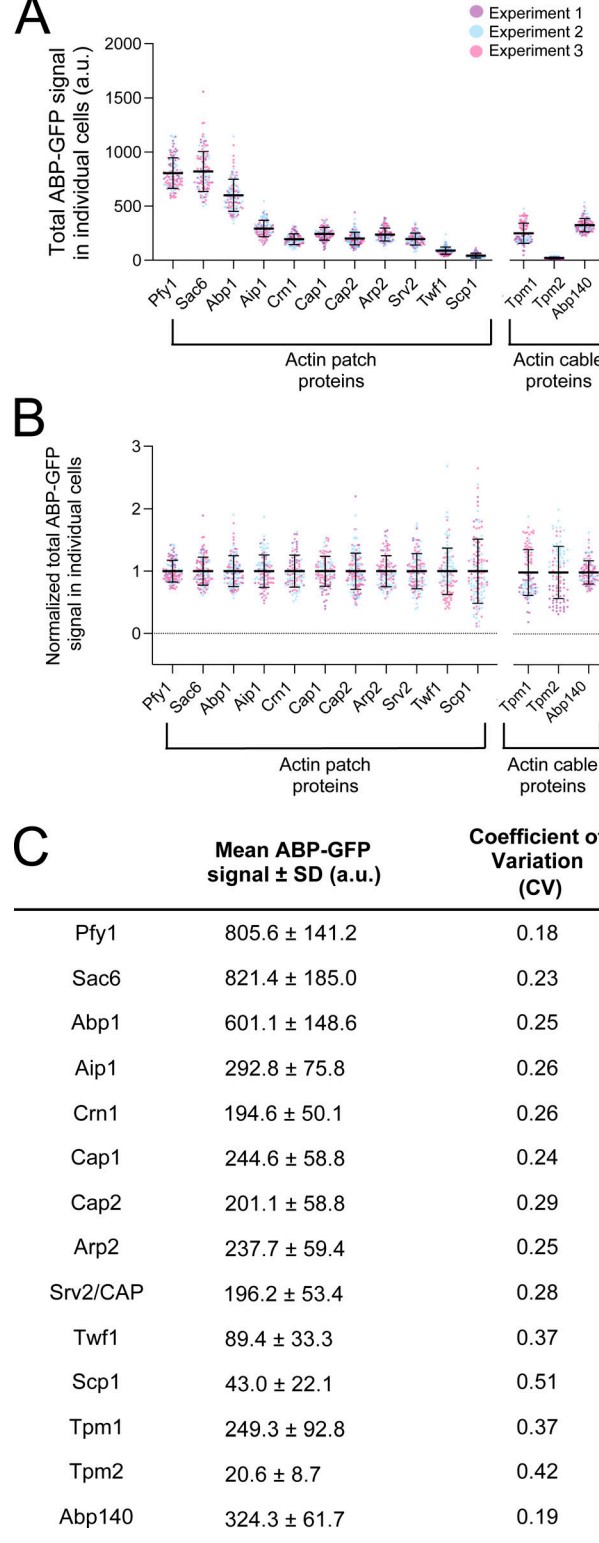

| | Mean ABP-GFP signal ± SD (a.u.) | Coefficient of Variation (CV) |
|---|---|---|
| Pfy1 | 805.6 ± 141.2 | 0.18 |
| Sac6 | 821.4 ± 185.0 | 0.23 |
| Abp1 | 601.1 ± 148.6 | 0.25 |
| Aip1 | 292.8 ± 75.8 | 0.26 |
| Crn1 | 194.6 ± 50.1 | 0.26 |
| Cap1 | 244.6 ± 58.8 | 0.24 |
| Cap2 | 201.1 ± 58.8 | 0.29 |
| Arp2 | 237.7 ± 59.4 | 0.25 |
| Srv2/CAP | 196.2 ± 53.4 | 0.28 |
| Twf1 | 89.4 ± 33.3 | 0.37 |
| Scp1 | 43.0 ± 22.1 | 0.51 |
| Tpm1 | 249.3 ± 92.8 | 0.37 |
| Tpm2 | 20.6 ± 8.7 | 0.42 |
| Abp140 | 324.3 ± 61.7 | 0.19 |

Figure S4.  **Cell-to-cell variance in ABP-GFP fluorescence levels. (A)** Total ABP-GFP signal for actin patch and cable proteins (mean ± SD). Data shown are from three independent experiments (*n* > 20 cells per experiment), with each data point representing one cell. Total number of cells analyzed for each ABP (left to right): 117, 116, 112, 96, 92, 99, 136, 102, 112, 113, 122, 98, 96, 102. Note that the actin cable proteins (Tpm1, Tpm2, and Abp140) were imaged on a different microscope from the patch proteins, which was better suited for imaging cables. Therefore, the intensities of the cable ABPs cannot be directly compared to the intensities of the patch ABPs. **(B)** Data from A normalized to the mean of each strain in order to more accurately assess variance. **(C)** Table showing the total fluorescence signal (mean ± SD) and the coefficient of variation (CV) for each ABP.

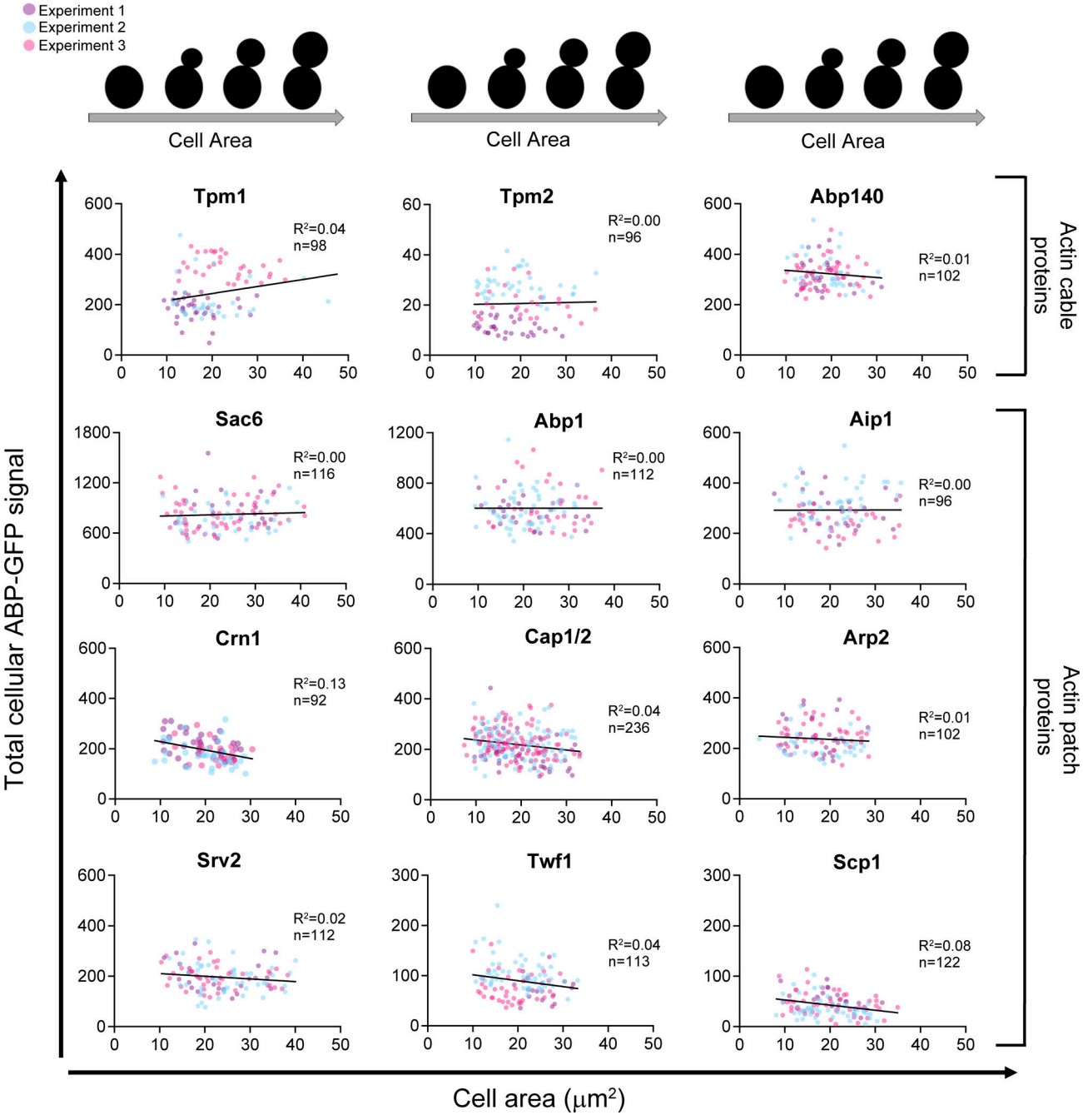

Figure S5.   **Total cellular levels of ABPs do not change significantly over the cell cycle.** For each ABP, the total GFP fluorescence value in a cell was plotted against the area (μm²) of the same cell. Then, linear regression analysis was used to correlate the total cellular concentration of ABP with increasing cell area (μm²) within the population.

**Provided online are Table S1, Table S2, and Table S3. Table S1 shows antibodies used for Western blots and the yeast actin binding proteins purified for generating standard curves. Table S2 shows comparison of the cytosolic fractions of five ABPs in *S. pombe* (Sirotkin et al., 2010) versus *S. cerevisiae* (this study). Table S3 shows *S. cerevisiae* strains used in this study.**

