## [Peer Review File · The Journal of Cell Biology]

Cytosolic concentrations of actin binding proteins and the implications for in vivo F-actin turnover

Sofia Gonzalez Rodriguez, Alison Wirshing, Anya Goodman, and Bruce Goode

Corresponding Author(s): Bruce Goode, Brandeis University

Review Timeline:

Submission Date:	2023-06-07
Editorial Decision:	2023-07-13
Revision Received:	2023-08-31
Editorial Decision:	2023-09-05
Revision Received:	2023-09-18

Monitoring Editor: Alex Mogilner

Scientific Editor: Andrea Marat

Transaction Report:

DOI: <https://doi.org/10.1083/jcb.202306036>

July 13, 2023

Re: JCB manuscript #202306036

Prof. Bruce L Goode
Brandeis University
Biology
Rosenstiel Center
415 South Street
Waltham, MA 02454

Dear Bruce,

Thank you for submitting your manuscript entitled "Quantitative inventory of *S. cerevisiae* actin cytoskeleton components leads to a model for how cells drive rapid F-actin turnover". The manuscript was assessed by expert reviewers, whose comments are appended to this letter. We invite you to submit a revision if you can address the reviewers' key concerns, as outlined here.

You will see that the reviewers are somewhat mixed in their assessment of your study. Editorially, we agree your study does not meet the criteria for an Article but find a suitably revised manuscript appropriate as a JCB Tool as we think it represents a potentially valuable resource for the community.

Guidelines for revising as a Tool:

- Provide some evidence to support the hypothesis that rapid actin turnover depends on high ABP content.
- Attempt to perform an analysis at the single-cell level to assess heterogeneity in the amount of different proteins from cell to cell.
- Re-examine the effect of cell size.
- Address the comments of reviewer 2 on quantification methodology (comments 3-5 + minor concerns).
- Address all minor concerns of reviewers 1 and 3.
- Concerns regarding cooperative binding and novelty (Reviewer 2 points 1 and 2) do not need to be addressed for a JCB Tool.

GENERAL GUIDELINES:

Text limits: Character count for an Article is < 40,000, not including spaces. Count includes title page, abstract, introduction, results, discussion, and acknowledgments. Count does not include materials and methods, figure legends, references, tables, or supplemental legends.

Figures: Articles may have up to 10 main text figures. Figures must be prepared according to the policies outlined in our Instructions to Authors, under Data Presentation, <https://jcb.rupress.org/site/misc/ifora.xhtml>. All figures in accepted manuscripts will be screened prior to publication.

Supplemental information: There are strict limits on the allowable amount of supplemental data. Articles may have up to 5 supplemental figures. Up to 10 supplemental videos or flash animations are allowed. A summary of all supplemental material should appear at the end of the Materials and methods section.

Please note that JCB now requires authors to submit Source Data used to generate figures containing gels and Western blots with all revised manuscripts. This Source Data consists of fully uncropped and unprocessed images for each gel/blot displayed in the main and supplemental figures. Since your paper includes cropped gel and/or blot images, please be sure to provide one Source Data file for each figure that contains gels and/or blots along with your revised manuscript files. File names for Source Data figures should be alphanumeric without any spaces or special characters (i.e., SourceDataF#, where F# refers to the associated main figure number or SourceDataFS# for those associated with Supplementary figures). The lanes of the gels/blots should be labeled as they are in the associated figure, the place where cropping was applied should be marked (with a box), and molecular weight/size standards should be labeled wherever possible.

The typical timeframe for revisions is three to four months. While most universities and institutes have reopened labs and allowed researchers to begin working at nearly pre-pandemic levels, we at JCB realize that the lingering effects of the COVID-19 pandemic may still be impacting some aspects of your work, including the acquisition of equipment and reagents. Therefore, if you anticipate any difficulties in meeting this aforementioned revision time limit, please contact us and we can work with you to find an appropriate time frame for resubmission. Please note that papers are generally considered through only one revision cycle, so any revised manuscript will likely be either accepted or rejected.

Thank you for this interesting contribution to Journal of Cell Biology. You can contact us at the journal office with any questions, cellbio@rockefeller.edu or call (212) 327-8588.

Sincerely,

Alex Mogilner
Monitoring Editor

Andrea L. Marat
Senior Scientific Editor

Journal of Cell Biology

Reviewer #1 (Comments to the Authors (Required)):

In this manuscript, Gonzalez Rodriguez et al. sought to measure cellular levels of actin and most actin-binding proteins present in *Saccharomyces cerevisiae*. They found that ABPs are mainly found in the cytosol and hypothesized that F-actin binding sites are limited in vivo. They propose that this mechanism could explain the rapid turnover of actin structures in the yeast cell.

The scientific question is of key importance and the approach is interesting, but further data will be needed to strengthen the manuscript and avoid certain conclusions being overstated on the basis of a limited number of supporting data.

More precisely, cytosolic concentrations have already been estimated a few years ago, in another strain (Sirotkin et al. MBoC 2010). What could be very interesting in the current manuscript is an approach at the single-cell level (which is somewhat done in Figure 4 but not commented on at all).

Therefore, to accept this paper for publication in J Cell Biol, I would recommend studying the data at the single-cell level and more specifically the effect of cell size, in order to be able to draw solid conclusions on this very important issue.

Major points

1. My first concern is the novelty of the article. What is really new in the manuscript is that the cytosolic concentration of ABPs is high. The authors hypothesize that this is the key parameter for rapid actin turnover, but this is not demonstrated. More evidence is needed to support this hypothesis, such as direct measurement of actin turnover as a function of ABP content.

2. Here, the calculation of protein quantity is performed at population level. But since the authors have fluorescently-labeled strains at their disposal, why not perform this analysis at the single-cell level to assess heterogeneity in the amount of different proteins from cell to cell? For ABPs that have been examined with a fluorescence signal, what is the variation from cell to cell? Are there proteins whose concentration is more constant than others? As the cytosolic concentration is measured in Figure 3, it should not be difficult to calculate the total concentration for each protein and display the results at single-cell level rather than population level.

3. Figure 4 shows how the percentage of protein in the cytosol depends on cell size. It shows that the signal in the cytosol is

constant as a function of cell size. However, what happens to the TOTAL signal is not shown. It is possible that the total amount increases with cell size. The conclusion in Figure 4 that "yeast ABP expression levels appear to be optimized to maintain high cytosolic concentrations of each ABP" seems overstated...

Moreover, the effect of cell size is not addressed at all in the discussion or the introduction. As a result, this section is currently under-used in the manuscript.

Minor points

1. For the 3 ABPs measured with the fluorescence intensity it would be nice to have the standard curve generated from the other data with the other proteins (the curve is mentioned in the methods but not shown).
2. In figure 2, what are the method's accuracy checks? Does this depend on the threshold value chosen? Is it the same threshold value for all proteins on cables or patches?
3. Figures 2 and 3 describe the same thing, and could be merged into a single figure.
4. In the discussion, the funneling effect is mentioned and given as an argument to explain how the cell can maintain a G-actin reservoir above the critical concentration. High cytosolic of capping protein to maintain high concentration of monomers: ok. But the funneling hypothesis is still very controversial and it is probably over claimed to say that this could explain how the large reservoir of monomers is maintained in the cytosol. If mentioned in the discussion alternative hypothesis should also be discussed.
5. I may have missed something with the protein name but no quantification of the amount of formins is presented in the data?

Reviewer #2 (Comments to the Authors (Required)):

This study quantifies the amount of actin and its partners in budding yeast, aiming at providing parameters for the future modeling analysis. The ratio between cytoplasmic and actin-bound fractions of each protein was also estimated based on the localization of GFP-tagged proteins along the cellular actin structures. The current manuscript doesn't seem to provide solid discussion based on the basic knowledge in actin biochemistry. In addition, the quantification methods need to be improved.

Major Points

- 1) Novelty should be evidenced by new quantitative modeling analysis with the obtained data. This study does not really extend our understanding sufficiently for publication in Journal of Cell Biology.
- 2) There is little discussion of how actin interactors might be regulated for the binding to F-actin. For example, cooperative binding is little considered. Cofilin and TPM bind F-actin cooperatively but not linearly in proportion to time and concentrations. Some molecules need to be activated (Arp2/3 complex) and require cofactors (Aip1). They do not share the same binding site on F-actin. In the case of Pfy1 and Cof1, the cytoplasmic concentration does not necessarily reflect the free concentration. These facts undermine the significance of measuring the concentration in the cytoplasm.
- 3) I am greatly concerned with the quantification methods. First, the estimation in the Methods, "(4) ~ 50% of the *S. cerevisiae* cell volume is cytoplasm, while the remaining 50% is occupied by cell wall and organelles (nucleus, vacuole, mitochondria, ER, Golgi, peroxisomes, etc.) (Uchida et al., 2011)" could be wrong. In the Uchida paper, they conclude that "the cytosol, other non-segmented organelles, large macromolecular complexes, such as the ribosomes, and the cytoskeleton occupy the remaining 70% of cell volume." Here we need to further consider that "70%" is against the true 3D volume of yeast cells and that this volume contains large macromolecules. Furthermore, there remains uncertainty regarding the difference between true cell volume and "wet weight" which may contain water in the extracellular space and how much of the cytoplasm is occupied by the free medium. I would predict 30 to 35% instead of "50%". The authors should measure their own samples for the number of cells and total amount of proteins, and more carefully estimate the volume of the free cytoplasmic space excluding macromolecular structures.
- 4) Second, the western blot procedure does not appear to sufficiently extract all proteins in yeast.
- 5) Third, the masking procedure is a bit sloppy in the live cell image analysis. The use of 2D maximum intensity projection images may not reliably partition the two locations. It is unacceptable to totally exclude the signals overlapping with the mask from the cytosolic fraction. This analysis needs improvement.

Minor Points

- 6) It is not appropriate to cite the previous quantitative study (Sirotkin et al, 2010) with a sentence, "Very few studies to date have measured the ..." (in the first Discussion paragraph, p7 bottom ~ p8) without introducing its key findings. In addition, cytosolic concentrations of actin and many regulators including capZ (Mol Biol Cell 1995 doi: 10.1091/mbc.6.12.1659), cofilin, profilin, Tbeta4, etc. have been measured previously.
- 7) In the text describing the western blot data, it should be mentioned that the values indicate "mean {plus minus} SD" as in the figure legends. In figures (or figure legends), the sample size should be noted for each western blot.

8) Do the western blot results and the GFP intensity data show good correlation?

9) In Figure 2, the images and the captions do not provide the identity of "ABP of interest". Include their identity. In particular, ABP of interest in Fig 2A appear to associate with non-actin structures too. Can any of the proteins localize to actin patches without binding F-actin? Discuss such mechanisms if any.

10) Regarding a sentence in p5, "Furthermore, our data reveal that Abp1 is much more abundant than each of the other ABPs to which it binds (Figure 1C), suggesting that there is enough Abp1 to bind each ligand separately and Abp1 does not necessarily form complexes simultaneously with all of these ligands.": This sentence is not meaningful without referring to relevant studies concerning different complexes of Abp1. If there is nothing to mention, it is better to delete this sentence. For example, if ligands of Abp1 form a stable complex, they would bind Abp1 simultaneously. On the other hand, it is generally expected that a variety of combinations are formed between equimolar molecules.

11) The estimated concentration of the profilin-actin complex in p5 is erratic. Actin elongation in cells might be slower than in vitro due to molecular crowding and slow diffusion in the cytoplasm. Free G-actin and cofilin-bound G-actin also need to be considered to estimate the G-actin concentration in the same paragraph. The last sentence in p5 is wrong accordingly.

12) The Discussion section can be shortened by removing re-interpretation of previous findings, many of which are not supported by nor related to the current data.

-

Reviewer #3 (Comments to the Authors (Required)):

The authors utilize a combination of quantitative western blotting and fluorescence microscopy to determine the cellular (and cytosolic) concentrations of actin and 14 ABPs in budding yeast. This manuscript represents a well-performed study that fills an important gap that is critical to the understanding of how yeast cells utilize a complicated mixture of ABPs to build functionally distinct F-actin networks. In my opinion this work will be of broad interest to the cytoskeleton community and those generally thinking about systems level cell biology problems.

I have some fairly minor comments to consider:

(1) My understanding is that tagging profilin with a fluorescent protein isn't straight forward, although it has been reported (Pimm et al., *Elife* 2022). A more detailed description of how Pfy1 was tagged endogenously with GFP should be included, as well as by what metrics the fusion (strain) was determined to be functional.

(2) Middle section page 6: I might have missed this, but the authors should describe how how tagged Tpm1 and Tpm2 expressed over the endogenous genes can be used to determine the endogenous concentrations of Tpm1 and Tpm2 in yeast cells.

(3) Table 1: Why is there not a value for the percent of Pfy1 in the cytosol? Furthermore, there appears to be an issue with the ** and *** designations at the bottom, and there is no description of *.

Reviewer #1 (Comments to the Authors (Required)):

In this manuscript, Gonzalez Rodriguez et al. sought to measure cellular levels of actin and most actin-binding proteins present in *Saccharomyces cerevisiae*. They found that ABPs are mainly found in the cytosol and hypothesized that F-actin binding sites are limited in vivo. They propose that this mechanism could explain the rapid turnover of actin structures in the yeast cell.

The scientific question is of key importance and the approach is interesting, but further data will be needed to strengthen the manuscript and avoid certain conclusions being overstated on the basis of a limited number of supporting data.

More precisely, cytosolic concentrations have already been estimated a few years ago, in another strain (Sirotkin et al. MBoC 2010). What could be very interesting in the current manuscript is an approach at the single-cell level (which is somewhat done in Figure 4 but not commented on at all).

Indeed, Sirotkin et al., 2010 studied AAP levels in fission yeast; however, *S. pombe* is separated by ~ 1 billion years of evolution from budding yeast *S. cerevisiae* (Sipiczki, 2000). More importantly, the Sirotkin study analyzed cytosolic levels for only 5 out of the 14 proteins we analyzed here; thus, our study provides new information on cytosolic levels for 9 new ABPs. Budding yeast is one of the main model organisms used to study actin dynamics regulation, and cytosolic concentrations for *S. cerevisiae* ABPs have not been available until now. Therefore, we feel strongly that our study fills an important gap, and will be very useful to the cytoskeleton community. We revised our Introduction to emphasize that the Sirotkin et al., 2010 study defined cytosolic levels of ABPs in a different system (fission yeast), and that our study focuses on budding yeast.

Therefore, to accept this paper for publication in J Cell Biol, I would recommend studying the data at the single-cell level and more specifically the effect of cell size, in order to be able to draw solid conclusions on this very important issue.

Major points

1. My first concern is the novelty of the article. What is really new in the manuscript is that the cytosolic concentration of ABPs is high. The authors hypothesize that this is the key parameter for rapid actin turnover, but this is not demonstrated. More evidence is needed to support this hypothesis, such as direct measurement of actin turnover as a function of ABP content.

The experimental support for this hypothesis comes from many previous studies, from our group and others, showing that mutations in these ABPs (e.g., *cof1-22*, *aip1Δ*, *crn1Δ*, *srv2Δ*) decrease rates of actin turnover in vivo (Balcer et al., 2003; Okada et al., 2006; Okreglak and Drubin, 2007; Gandhi et al., 2009). These studies provide clear support for this hypothesis. Further tests of this hypothesis (that high cytosolic levels of ABPs are required for rapid actin turnover) are challenging to design, because lowering the levels of any individual ABP can alter the levels of the remaining ABPs. More conclusively testing this hypothesis will require future efforts to genetically disrupt each individual ABP and determine how this alters the cytosolic levels of all the remaining

ABPs, an enormous experimental endeavor that is obviously beyond the scope of the current study. Finally, we don't see the harm in introducing the hypothesis above, as a working model, since it arises from our findings. We have emphasized that this model is only a working model, and awaits further validation.

2. Here, the calculation of protein quantity is performed at population level. But since the authors have fluorescently-labeled strains at their disposal, why not perform this analysis at the single-cell level to assess heterogeneity in the amount of different proteins from cell to cell? For ABPs that have been examined with a fluorescence signal, what is the variation from cell to cell? Are there proteins whose concentration is more constant than others? As the cytosolic concentration is measured in Figure 3, it should not be difficult to calculate the total concentration for each protein and display the results at single-cell level rather than population level.

Following the reviewer's suggestion, we have added new graphs comparing the cell-to-cell variance in total ABP-GFP signal for each ABP (Supplemental Figure 4A and 4B). Additionally, we have added a table that includes the coefficient of variation (CV) for each ABP (Supplemental Figure 4C). These data reveal low variance among cells in ABP-GFP levels.

3. Figure 4 shows how the percentage of protein in the cytosol depends on cell size. It shows that the signal in the cytosol is constant as a function of cell size. However, what happens to the TOTAL signal is not shown. It is possible that the total amount increases with cell size. The conclusion in Figure 4 that "yeast ABP expression levels appear to be optimized to maintain high cytosolic concentrations of each ABP" seems overstated...

Following the reviewer's suggestion, we performed this analysis (Supplemental Figure 5), which shows that total ABP-GFP levels, like cytosolic ABP-GFP levels, do not change significantly over the cell cycle.

Moreover, the effect of cell size is not addressed at all in the discussion or the introduction. As a result, this section is currently under-used in the manuscript.

We had not discussed this topic because we had not observed significant changes in ABP levels with cell size. To address this, we now mention in the Results that these observations are consistent with previous yeast proteomic studies showing that the concentrations of most proteins in the cell remain constant even as cells increase in size (Lin and Amir, 2018).

Minor points

1. For the 3 ABPs measured with the fluorescence intensity it would be nice to have the standard curve generated from the other data with the other proteins (the curve is mentioned in the methods but not shown).

As suggested, we now show the standard curve (Supplemental Figure 1D).

2. In figure 2, what are the method's accuracy checks? Does this depend on the threshold value chosen? Is it the same threshold value for all proteins on cables or patches?

For measuring cytosolic (unmasked) concentrations of ABP-GFPs, we subtracted background fluorescence, which was consistently measured from untagged cells grown and imaged in parallel for each experiment. For selecting where to outline the patch and cable masks, we applied the same threshold value to all cells (different for patches versus cables). As an additional accuracy check, we have now reanalyzed the same cell images using three different threshold values (our normal 'set point', as well as 25% lower and 25% higher thresholds), and obtained similar results for ABP-GFP levels (this is shown in a new panel: Figure 2C).

3. Figures 2 and 3 describe the same thing, and could be merged into a single figure. We have added new data to Figure 2 (see minor point #2 above). This would make it difficult to merge these figures without compromising the visibility of the data, and since the JCB Tool format allows up to 10 main figures, and we only currently have 5 main figures (plus one table), we prefer to keep these figures separate.

4. In the discussion, the funneling effect is mentioned and given as an argument to explain how the cell can maintain a G-actin reservoir above the critical concentration. High cytosolic of capping protein to maintain high concentration of monomers: ok. But the funneling hypothesis is still very controversial and it is probably over claimed to say that this could explain how the large reservoir of monomers is maintained in the cytosol. If mentioned in the discussion alternative hypothesis should also be discussed. We have revised our Discussion to provide more details, and say that the monomer pool likely builds up as a result of multiple factors, including high levels of profilin to bind monomers, high levels of capping protein to rapidly cap barbed ends of filaments, and tight spatiotemporal restriction on actin nucleation events in vivo. However, this essentially is the 'funneling effect', and therefore believe that this warrants referencing it.

5. I may have missed something with the protein name but no quantification of the amount of formins is presented in the data? We did not include formins in this study because they are extremely low abundance in cells (estimated ~100 dimers per cell, translating to ~ 10 nM), difficult to detect, and difficult to quantify free versus F-actin bound levels. Our focus here was on the abundant ABPs, and how their levels create competition for limited binding sites on F-actin. We have recently published a study that addresses the relationship between formins and capping protein and how this influences yeast actin organization and dynamics (Wirshing et al., 2023 JCB).

Reviewer #2 (Comments to the Authors (Required)):

This study quantifies the amount of actin and its partners in budding yeast, aiming at providing parameters for the future modeling analysis. The ratio between cytoplasmic and actin-bound fractions of each protein was also estimated based on the localization of GFP-tagged proteins along the cellular actin structures. The current manuscript doesn't seem to provide solid discussion based on the basic knowledge in actin

biochemistry. In addition, the quantification methods need to be improved.

Major Points

1) Novelty should be evidenced by new quantitative modeling analysis with the obtained data. This study does not really extend our understanding sufficiently for publication in *Journal of Cell Biology*.

2) There is little discussion of how actin interactors might be regulated for the binding to F-actin. For example, cooperative binding is little considered. Cofilin and TPM bind F-actin cooperatively but not linearly in proportion to time and concentrations. Some molecules need to be activated (Arp2/3 complex) and require cofactors (Aip1). They do not share the same binding site on F-actin. In the case of Pfy1 and Cof1, the cytoplasmic concentration does not necessarily reflect the free concentration. These facts undermine the significance of measuring the concentration in the cytoplasm.

3) I am greatly concerned with the quantification methods. First, the estimation in the Methods, "(4) ~ 50% of the *S. cerevisiae* cell volume is cytoplasm, while the remaining 50% is occupied by cell wall and organelles (nucleus, vacuole, mitochondria, ER, Golgi, peroxisomes, etc.) (Uchida et al., 2011)" could be wrong. In the Uchida paper, they conclude that "the cytosol, other non-segmented organelles, large macromolecular complexes, such as the ribosomes, and the cytoskeleton occupy the remaining 70% of cell volume." Here we need to further consider that "70%" is against the true 3D volume of yeast cells and that this volume contains large macromolecules. Furthermore, there remains uncertainty regarding the difference between true cell volume and "wet weight" which may contain water in the extracellular space and how much of the cytoplasm is occupied by the free medium. I would predict 30 to 35% instead of "50%". The authors should measure their own samples for the number of cells and total amount of proteins, and more carefully estimate the volume of the free cytoplasmic space excluding macromolecular structures.

The Uchida estimate (of 50% volume being cytosolic) is the best available. Further, we did our own estimates (see Methods) and they agree with the Uchida study. Importantly, even if the values turn out to be off by 10-20%, in either direction, this does not have a major impact on the conclusion of our findings, that the fraction of ABP in the cytosol (versus bound to F-actin structures) is high, and remains high over the cell cycle. It would also not change the relative abundances of the ABPs to each other.

4) Second, the western blot procedure does not appear to sufficiently extract all proteins in yeast.

We do not understand the basis for this claim. We stated in the Methods that only 1 out of the 12 ABPs we analyzed by Western blotting (Aip1) had issues with not being fully extracted by the method used for all others, and therefore we used a different extraction procedure for Aip1, which was fully effective in extracting Aip1 from the cell.

5) Third, the masking procedure is a bit sloppy in the live cell image analysis. The use of 2D maximum intensity projection images may not reliably partition the two locations. It is

unacceptable to totally exclude the signals overlapping with the mask from the cytosolic fraction. This analysis needs improvement.

We did not use the masking to determine the signal (ABP-GFP) associated with F-actin structures, but rather to assess the concentration (mean intensity) of ABP-GFP in areas *outside* of the F-actin masked structures. Therefore, as long as our analysis does a good job of excluding the F-actin patches, or cables, it will reliably report the concentration of the remaining (cytosolic) signal. We have clarified this point in the Results. In addition, we added new data (Figure 2C) demonstrating the robustness our approach for measuring the amount of protein in the cytosol.

Minor Points

6) It is not appropriate to cite the previous quantitative study (Sirotkin et al, 2010) with a sentence, "Very few studies to date have measured the ..." (in the first Discussion paragraph, p7 bottom ~ p8) without introducing its key findings. In addition, cytosolic concentrations of actin and many regulators including capZ (Mol Biol Cell 1995 doi: 10.1091/mbc.6.12.1659), cofilin, profilin, Tbeta4, etc. have been measured previously. We now discuss the findings of Sirotkin upon first citation, and cite studies from Zigmond and colleagues measuring cytosolic levels of capping protein, profilin and thymosin- β 4 in neutrophils (Cassimeris et al., 1992 JCB; DiNubile et al., 1995).

7) In the text describing the western blot data, it should be mentioned that the values indicate "mean {plus minus} SD" as in the figure legends. In figures (or figure legends), the sample size should be noted for each western blot.

We have introduced these changes to the text (mean +/- SD). Sample sizes for each Western blot are listed in Table 1. We have also added them to the legend for Figure 1, with a callout to Table 1.

8) Do the western blot results and the GFP intensity data show good correlation?

To address this, we added a new figure (Supplementary Figure 2A and 2B) comparing the western and GFP intensity data, and discuss this in the manuscript. This shows that for many ABPs there is good agreement, and in the few instances where the data are not in good agreement, we offer potential explanations.

9) In Figure 2, the images and the captions do not provide the identity of "ABP of interest". Include their identity. In particular, ABP of interest in Fig 2A appear to associate with non-actin structures too. Can any of the proteins localize to actin patches without binding F-actin? Discuss such mechanisms if any.

These data are for Sac6-GFP and mNeon-Tpm1, which we now label in the figure itself. The reviewer is correct, some yeast ABPs can be recruited to F-actin structures in vivo independent of their direct physical interactions with F-actin (e.g., twinfilin can be recruited to patches via its interactions with capping protein, and Srv2/CAP can be recruited to patches via its interactions with Abp1). While this does not change our conclusions about what fraction of these proteins are cytosolic, it may influence models and thinking about competition for binding sites on F-actin. We mention this in the 'Limitations to this Study' section, found before the Methods.

10) Regarding a sentence in p5, "Furthermore, our data reveal that Abp1 is much more abundant than each of the other ABPs to which it binds (Figure 1C), suggesting that there is enough Abp1 to bind each ligand separately and Abp1 does not necessarily form complexes simultaneously with all of these ligands.": This sentence is not meaningful without referring to relevant studies concerning different complexes of Abp1. If there is nothing to mention, it is better to delete this sentence. For example, if ligands of Abp1 form a stable complex, they would bind Abp1 simultaneously. On the other hand, it is generally expected that a variety of combinations are formed between equimolar molecules.

Agreed. We have deleted this claim.

11) The estimated concentration of the profilin-actin complex in p5 is erratic. Actin elongation in cells might be slower than in vitro due to molecular crowding and slow diffusion in the cytoplasm. Free G-actin and cofilin-bound G-actin also need to be considered to estimate the G-actin concentration in the same paragraph. The last sentence in p5 is wrong accordingly.

We tried to be clear in stating that this number is only an estimate (the best estimate we can make with the data available to us). We agree that there are many conditions and factors in cells that could increase or decrease cable extension rates. As these factors are discovered, the interpretation of the numbers we provide here should be revisited and reconsidered in light of those findings. However, there is merit to having a Discussion that leans on these estimates, as it reflects our current understanding in the field.

12) The Discussion section can be shortened by removing re-interpretation of previous findings, many of which are not supported by nor related to the current data.

We made a concerted effort to address this point, and have edited and shortened the Discussion.

Reviewer #3 (Comments to the Authors (Required)):

The authors utilize a combination of quantitative western blotting and fluorescence microscopy to determine the cellular (and cytosolic) concentrations of actin and 14 ABPs in budding yeast. This manuscript represents a well-performed study that fills an important gap that is critical to the understanding of how yeast cells utilize a complicated mixture of ABPs to build functionally distinct F-actin networks. In my opinion this work will be of broad interest to the cytoskeleton community and those generally thinking about systems level cell biology problems.

I have some fairly minor comments to consider:

(1) My understanding is that tagging profilin with a fluorescent protein isn't straight forward, although it has been reported (Pimm et al., Elife 2022). A more detailed

description of how Pfy1 was tagged endogenously with GFP should be included, as well as by what metrics the fusion (strain) was determined to be functional.

Profilin levels were determined by GFP fluorescence using the standard curve shown in Supplemental Figure 1D and as described in Methods. Our C-terminally GFP-tagged profilin (Pfy1-GFP) is entirely cytoplasmic, as expected and similar to untagged profilin. We also show that the Pfy1-GFP strain grows similar to wildtype at 25, 34, and 37°C (Supplemental Figure 1E), suggesting that Pfy1-GFP is functional.

(2) Middle section page 6: I might have missed this, but the authors should describe how how tagged Tpm1 and Tpm2 expressed over the endogenous genes can be used to determine the endogenous concentrations of Tpm1 and Tpm2 in yeast cells.

Thank you. We have clarified this in the Results. The cellular concentration of endogenous (untagged) Tpm1 was first determined by quantitative Western blotting (12.4 μ M). Next, since we did not have an antibody to Tpm2, we used knowledge from an earlier study showing there is a 6:1 molar ratio of Tpm1 to Tpm2 in cells to estimate the cellular concentration of Tpm2 (2.54 μ M). The mNeon-tagged constructs of Tpm1 and Tpm2 were only used to determine what fraction of each protein is cytosolic versus associated with F-actin cables. The one assumption this makes is that the tagged constructs distribute themselves (between F-actin and cytosol) similar to the untagged proteins. We have no way of verifying this assumption, but we mention this caveat in the 'Limitations of this study' section found just before the Methods.

(3)Table 1: Why is there not a value for the percent of Pfy1 in the cytosol? Furthermore, there appears to be an issue with the ** and *** designations at the bottom, and there is no description of *.

There is no percentage of Pfy1 in the cytosol because it is entirely cytosolic, i.e., it does not associate with F-actin. Designations of *, **, and *** have been fixed. Thank you.

September 5, 2023

RE: JCB Manuscript #202306036R

Prof. Bruce L Goode
Brandeis University
Biology
Rosenstiel Center
415 South Street
Waltham, MA 02454

Dear Bruce,

Thank you for submitting your revised manuscript entitled "Cytosolic concentrations of actin binding proteins and implications for in vivo F-actin turnover". We would be happy to publish your paper in JCB pending final revisions necessary to meet our formatting guidelines (see details below).

A. MANUSCRIPT ORGANIZATION AND FORMATTING:

- 1) Text limits: Character count for Tools is < 40,000, not including spaces. Count includes abstract, introduction, results, discussion, and acknowledgments. Count does not include title page, figure legends, materials and methods, references, tables, or supplemental legends.
- 2) Figures limits: Tools may have up to 10 main text figures.
- 3) Figure formatting: Scale bars must be present on all microscopy images, including inset magnifications. Molecular weight or nucleic acid size markers must be included on all gel electrophoresis.
- 4) Statistical analysis: Error bars on graphic representations of numerical data must be clearly described in the figure legend. The number of independent data points (n) represented in a graph must be indicated in the legend. Statistical methods should be explained in full in the materials and methods. For figures presenting pooled data the statistical measure should be defined in the figure legends. Please also be sure to indicate the statistical tests used in each of your experiments (either in the figure legend itself or in a separate methods section) as well as the parameters of the test (for example, if you ran a t-test, please indicate if it was one- or two-sided, etc.). Also, if you used parametric tests, please indicate if the data distribution was tested for normality (and if so, how). If not, you must state something to the effect that "Data distribution was assumed to be normal but this was not formally tested."
- 5) Abstract and title: The abstract should be no longer than 160 words and should communicate the significance of the paper for a general audience. The title should be less than 100 characters including spaces. Make the title concise but accessible to a general readership.
- 6) Materials and methods: Should be comprehensive and not simply reference a previous publication for details on how an experiment was performed. Please provide full descriptions in the text for readers who may not have access to referenced manuscripts.
- 7) Please be sure to provide the sequences for all of your primers/oligos and RNAi constructs in the materials and methods. You must also indicate in the methods the source, species, and catalog numbers (where appropriate) for all of your antibodies. Please also indicate the acquisition and quantification methods for immunoblotting/western blots.
- 8) Microscope image acquisition: The following information must be provided about the acquisition and processing of images:
 - a. Make and model of microscope
 - b. Type, magnification, and numerical aperture of the objective lenses
 - c. Temperature
 - d. Imaging medium
 - e. Fluorochromes
 - f. Camera make and model

g. Acquisition software

h. Any software used for image processing subsequent to data acquisition. Please include details and types of operations involved (e.g., type of deconvolution, 3D reconstitutions, surface or volume rendering, gamma adjustments, etc.).

10) Supplemental materials: There are strict limits on the allowable amount of supplemental data. Articles may have up to 5 supplemental figures. Please also note that tables, like figures, should be provided as individual, editable files. A summary of all supplemental material should appear at the end of the Materials and methods section.

13) ORCID IDs: ORCID IDs are unique identifiers allowing researchers to create a record of their various scholarly contributions in a single place. Please note that ORCID IDs are now *required* for all authors. At resubmission of your final files, please be sure to provide your ORCID ID and those of all co-authors.

Please note that JCB now requires authors to submit Source Data used to generate figures containing gels and Western blots with all revised manuscripts. This Source Data consists of fully uncropped and unprocessed images for each gel/blot displayed in the main and supplemental figures. Since your paper includes cropped gel and/or blot images, please be sure to provide one Source Data file for each figure that contains gels and/or blots along with your revised manuscript files. File names for Source Data figures should be alphanumeric without any spaces or special characters (i.e., SourceDataF#, where F# refers to the associated main figure number or SourceDataFS# for those associated with Supplementary figures). The lanes of the gels/blots should be labeled as they are in the associated figure, the place where cropping was applied should be marked (with a box), and molecular weight/size standards should be labeled wherever possible.

Journal of Cell Biology now requires a data availability statement for all research article submissions. These statements will be published in the article directly above the Acknowledgments. The statement should address all data underlying the research presented in the manuscript. Please visit the JCB instructions for authors for guidelines and examples of statements at (<https://rupress.org/jcb/pages/editorial-policies#data-availability-statement>).

B. FINAL FILES:

**It is JCB policy that if requested, original data images must be made available to the editors. Failure to provide original images upon request will result in unavoidable delays in publication. Please ensure that you have access to all original data images prior

to final submission.**

Thank you for this interesting contribution, we look forward to publishing your paper in Journal of Cell Biology.

Sincerely,

Alex Mogilner
Monitoring Editor

Andrea L. Marat
Senior Scientific Editor

Journal of Cell Biology